# Autobidding Auctions with LLM-Powered Creatives

**Bingzhe Wang** [* 1]  **Bowei Zhang** [* 1]  **Changyuan Yu** [2]  **Qi Qi** [† 1 3 4]

## Abstract

The integration of Large Language Models (LLMs) into ad auctions for dynamic creative enhancement presents a paradigm shift, yet introduces significant computational costs disrupting traditional mechanism design. This paper provides a comprehensive game-theoretic and algorithmic framework for such LLM-augmented auctions. We model the system as a dynamic Stackelberg game where the platform (leader) strategically invests in creative enhancement to maximize net revenue, while autobidding agents (followers) respond to enhanced ad qualities under budget constraints. To endogenize inference costs, we propose the Platform-Investment Mechanism (PIM). We develop the Online Dual-Descent Bidding with Regularization (ODDB-R) algorithm for agents to learn optimal bidding strategies in this non-stationary environment. For the platform, we formulate the investment problem as a continuous control task and solve it using a Two-Timescale Stackelberg Learning with Proximal Policy Optimization (TTSL-PPO) algorithm, which provably converges to a Stackelberg Stationary Point. Extensive experiments on large-scale real-world datasets and state-of-the-art LLMs demonstrate that our framework significantly outperforms heuristic baselines in revenue, social welfare, and user engagement.

## 1. Introduction

The digital advertising ecosystem is undergoing a paradigm shift driven by two forces: the ubiquity of automated bidding (autobidding) and the rapid ascent of Large Language Models (LLMs). In modern real-time auctions, advertisers define high-level objectives—such as maximizing conversions subject to budget constraints—which automated agents translate into bids in real-time (Aggarwal et al., 2019). Ad campaign efficacy is increasingly determined by creative quality, where superior creatives directly elevate Click-Through Rates (CTR), enhancing impression value and driving market efficiency.

Traditionally labor-intensive, creative design can now be automated via LLMs (Cai et al., 2025; Sun et al., 2024). While recent research explores generating creatives from scratch (Duetting et al., 2024; Feizi et al., 2023; Soumalias et al., 2026), such approaches face practical hurdles. Generating long-form content is computationally expensive and slow, failing to meet the strict latency requirements of real-time auctions. Additionally, these methods often prioritize generation quality in a vacuum, neglecting user-specific context and the advertiser's brand voice.

We propose a pragmatic, scalable paradigm: *LLM-based creative enhancement*. Rather than generating new ads from zero, the platform leverages LLMs to dynamically refine existing ad *titles* based on user query context. This ensures low latency, high relevance, and adherence to the advertiser's message. We model this enhancement as a strategic **investment**. Since LLM inference incurs non-negligible costs (e.g., API fees), the platform must decide *ex-ante* on enhancement investments. This investment is risky: the platform bears the cost immediately to increase potential ad value, hoping to recoup expenditure through higher clearing prices. Crucially, to reconcile costs with real-time constraints, we employ a predict-then-execute workflow: agents bid based on *predicted* enhanced metrics, while the costly LLM inference is executed only for the winning ad.

We formalize this interaction as a Stackelberg game. The platform (leader) commits to an investment policy via a novel mechanism: the **Platform-Investment Mechanism (PIM)**. In PIM, the platform strategically allocates resources to enhance ad titles, maximizing *net* revenue (payments minus inference costs). Autobidding agents (followers) observe enhanced qualities and adjust bids to maximize value under budget constraints. We analyze this game within a Generalized Second-Price (GSP) auction setting (Edelman

[*]Equal contribution  [1]Gaoling School of Artificial Intelligence, Renmin University of China, Beijing, China  [2]Baidu Inc., Beijing, China  [3]Beijing Key Laboratory of Research on Large Models and Intelligent Governance, Beijing, China  [4]Engineering Research Center of Next-Generation Intelligent Search and Recommendation, MOE, Beijing, China. Correspondence to: Qi Qi <qi.qi@ruc.edu.cn>.

*Proceedings of the 43rd International Conference on Machine Learning*, Seoul, South Korea. PMLR 306, 2026. Copyright 2026 by the author(s).

et al., 2007).

By endogenizing creative enhancement costs, the PIM-induced subgame preserves the essential equilibrium properties of the original autobidding system. Moreover, the mechanism acts as a modular wrapper around existing auctions, allowing PIM to be deployed as an *incremental modification* to complex ad architectures, avoiding the engineering risks of full-scale overhauls.

Optimizing this policy is non-trivial due to two challenges: the subgame among budget-constrained autobidders admits multiple Nash Equilibria (Conitzer et al., 2022), rendering the leader's objective ill-defined; and the environment is non-stationary, requiring concurrent learning from censored feedback. To address these, our main contributions are:

(1) **Equilibrium Selection via Regularization:** To resolve the stability risk posed by multiple Pacing Equilibria, we introduce a regularized game formulation. We prove this game possesses a unique Pure Strategy Nash Equilibrium (PNE) (Lemma 3.3) and show it closely approximates the original game (Theorem 3.4), providing a stable optimization target.

(2) **Agent Learning Algorithm:** For the dynamic agent subgame, we develop the *Online Dual-Descent Bidding with Regularization (ODDB-R)* algorithm. We prove that under ODDB-R, agents' joint strategies converge to the unique regularized equilibrium (Theorem 4.1). We further establish an individual $O(\sqrt{T})$ no-regret guarantee against the optimal offline benchmark (Theorem 4.2).

(3) **Concurrent Stackelberg Learning:** We tackle the bilevel optimization by designing the *Two-Timescale Stackelberg Learning with Proximal Policy Optimization (TTSL-PPO)* algorithm. This method couples agents' fast-timescale bidding updates with the platform's slow-timescale policy optimization. We provide a theoretical analysis proving the convergence of the algorithm (Theorem 5.2).

(4) **Extensive Empirical Validation:** We conduct extensive evaluations using the large-scale *AntM2C* dataset (Huan et al., 2024) with *GPT-4o*, verifying performance under authentic generative AI conditions. Results demonstrate PIM significantly increases platform revenue and social welfare compared to baselines, confirming the economic viability of LLM-powered creative enhancement.

### 1.1. Related Work

Our research builds upon foundational online ad auction theory, specifically Generalized First-Price (GFP) (Despotakis et al., 2021) and Generalized Second-Price (GSP) (Edelman et al., 2007) mechanisms. The shift towards autobidding has spurred extensive research into optimizing bidding algorithms (Lu et al., 2017; Zhou et al., 2018; Liang et al., 2023), characterizing equilibria in constrained settings (Ag-garwal et al., 2019; Chen et al., 2024; Aggarwal et al., 2023; Conitzer et al., 2022; Yan et al., 2026; Huang et al., 2026), and designing mechanisms for value-maximizing agents (Balseiro et al., 2021; Golrezaei et al., 2021; Balseiro et al., 2023).

Integrating LLMs into economic systems is an expanding frontier. One strand utilizes LLMs as proxy agents to investigate phenomena like algorithmic collusion (Agrawal et al., 2025), negotiation dynamics (Xia et al., 2024), and preference elicitation (Huang et al., 2025). While promising, the alignment of LLM behavior with human rationality remains debated (Shah et al., 2025; Zhu et al.; Tsuchihashi, 2023). A second stream explores LLMs in mechanism design, including auctioning tokens for generative content (Duetting et al., 2024), summarizing information for allocation (Dubey et al., 2024), and empowering personalized valuations (Cai et al., 2025; Sun et al., 2024; Bergemann et al., 2025). In advertising, recent works apply LLMs to generate creatives (Feizi et al., 2023; Soumalias et al., 2026), optimize sponsored search (Reisenbichler et al., 2025), or place ads via Retrieval-Augmented Generation (Hajiaghayi et al., 2024). Recent approaches like *LLM-Auction* (Zhao et al., 2025) and *ACQ* (Wang et al., 2024) also integrate generative capabilities. We differentiate by focusing on *creative enhancement* of titles—a latency-sensitive task feasible for real-time deployment—and rigorously endogenizing associated costs within a Stackelberg game (Jamshidi et al., 2024).

## 2. Model and Preliminaries

We model the online advertising ecosystem as a Stackelberg game between the platform (the *leader*) and $I$ autobidding agents (the *followers*) over a finite horizon of $T$ rounds, indexed by $\mathcal{T} = \{1, \ldots, T\}$. In each round, the platform auctions a single ad slot. Our framework's core innovation is integrating LLMs to dynamically enhance ad creatives (e.g., refining ad titles). We model the computational expense of LLM inference as a strategic investment determined by the platform. The objective of this investment is to elevate an ad's quality—primarily characterized by its CTR—transforming the baseline metric $\alpha_{ti}$ into a superior, enhanced metric $\hat{\alpha}_{ti}$. The platform determines investment levels based on the original ad title and observable user features (e.g., search history). In contrast, agents, lacking global information and computational resources, cannot initiate enhancements themselves. To formalize this, we first define the system's operational flow, denoted as the Platform-Investment Mechanism, and subsequently detail the optimization problems faced by agents and the platform.

### 2.1. The Platform-Investment Mechanism (PIM)

The interaction proceeds sequentially through five stages in each round or over the horizon, as illustrated in Figure 1:

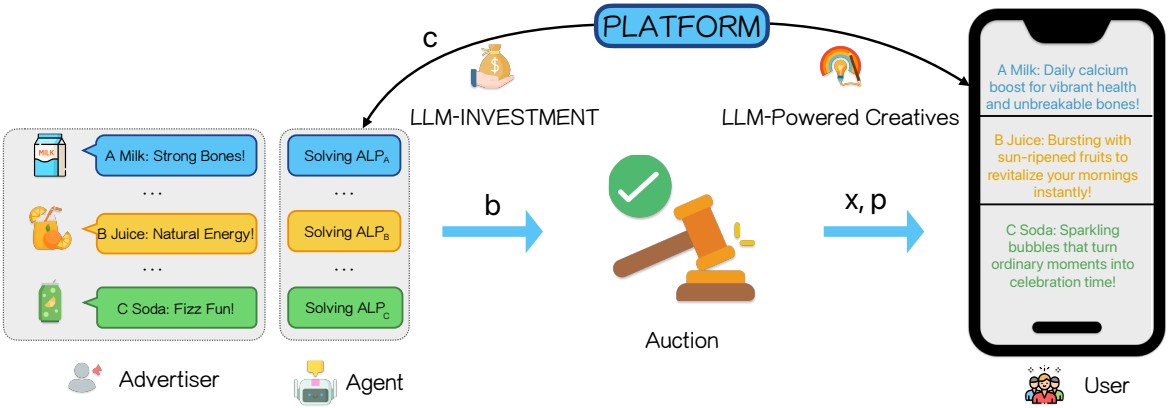

*Figure 1.* Overall framework of the Platform-Investment Mechanism (PIM).

(1) **Investment Decision:** The platform operates under a dynamic investment policy $\pi$. At the beginning of each round $t \in \mathcal{T}$, based on the current state (including original ad title, observable user features, and agent budgets), the platform determines an investment vector $\boldsymbol{c}^t = (c_{t1}, \dots, c_{tI}) \in \mathbb{R}_{\geq 0}^I$. Let $\boldsymbol{C} = \{\boldsymbol{c}^t\}_{t \in \mathcal{T}}$ denote cumulative investment decisions over the horizon.

(2) **Creative Enhancement Prediction:** For each agent $i$, investment $c_{ti}$ quantifies the computational effort (e.g., token length) allocated to the LLM. To resolve the latency and cost of running LLMs for all candidates, the platform first *predicts* the enhancement outcome. It transforms the baseline CTR via a deterministic, non-decreasing, and concave estimator function $\phi$, yielding the predicted enhanced metric $\hat{\alpha}_{ti} = \phi(\alpha_{ti}, c_{ti})$. Agents observe this predicted $\hat{\alpha}_{ti}$ to formulate their bids. The value-per-click $v_{ti}$ remains exogenous.[1]

(3) **Auction (GSP):** Agents observe the predicted enhanced CTR $\hat{\alpha}_{ti}$ and their valuation $v_{ti}$, and participate in a Generalized Second-Price (GSP) auction. The auction proceeds as follows: *(i) Bidding:* Each agent $i$ submits a per-click bid $b_{ti} \geq 0$. Let $\boldsymbol{b}^t = (b_{t1}, \dots, b_{tI})$ denote the bid vector. *(ii) Scoring:* The platform computes rank score $e_{ti} = \hat{\alpha}_{ti} \cdot b_{ti}$. *(iii) Allocation:* The slot is awarded to the agent with the highest score, $i^*(t) = \arg\max_{i \in \mathcal{I}}\{e_{ti}\}$. Let $x_{ti}(\boldsymbol{b}^t) \in \{0, 1\}$ indicate if agent $i$ wins (ties broken uniformly). *(iv) Payment:* The winner $i^*(t)$ pays the second-highest score $e_t^{(2)}(\boldsymbol{b}^t) = \max_{j \neq i^*(t)}\{e_{tj}\}$; losers pay zero.

(4) **Execution and Cost Incurrence:** Crucially, the platform executes the actual, costly LLM inference (and thus incurs the investment cost $c_{ti}$) *only* for the winning agent $i$ to generate the final creative for display ($x_{ti}(\boldsymbol{b}^t) = 1$). This *predict-then-execute* workflow aligns incentives while minimizing system overhead.

(5) **Payoff Realization:** The winner derives value $v_{ti}\hat{\alpha}_{ti}$ from the impression. The platform collects payment $e_t^{(2)}(\boldsymbol{b}^t)$ minus cost $c_{ti}$.

### 2.2. Agent Strategy and Equilibrium

We now analyze autobidding agent behavior within the subgame induced by the platform's investment decisions. Let $\mathcal{G}(\boldsymbol{C})$ denote the follower's game given enhanced metrics $\{\hat{\alpha}_{ti}\}_{i \in \mathcal{I}, t \in \mathcal{T}}$ resulting from $\boldsymbol{C}$. Agents are value-maximizers constrained by hard budgets. An agent $i$'s strategy is a sequence of bids $\boldsymbol{b}_i = (b_{it})_{t \in \mathcal{T}}$. Given competitors' strategies $\boldsymbol{b}_{-i}$, agent $i$ solves:

$$\max_{\boldsymbol{b}_i \in \mathbb{R}_{\geq 0}^T} \quad U_i(\boldsymbol{b}_i, \boldsymbol{b}_{-i}) = \sum_{t \in \mathcal{T}} \mathbb{E}_{\boldsymbol{b}_{-i}}[x_{ti}(\boldsymbol{b}^t) \cdot v_{ti} \cdot \hat{\alpha}_{ti}] \quad (1)$$

$$\text{s.t.} \quad \sum_{t \in \mathcal{T}} \mathbb{E}_{\boldsymbol{b}_{-i}}[x_{ti}(\boldsymbol{b}^t) \cdot e_t^{(2)}(\boldsymbol{b}^t)] \leq B_i \quad (2)$$

where $U_i$ is agent $i$'s utility, $B_i$ is the budget, and expectation $\mathbb{E}_{\boldsymbol{b}_{-i}}$ accounts for stochasticity in competitor bids and tie-breaking. We distinguish between exogenous parameters (baseline CTR $\alpha_{ti}$, value $v_{ti}$) and endogenous parameters (enhanced CTR $\hat{\alpha}_{ti}$), the latter being the outcome of the platform's strategy. We focus on the Pure Strategy Nash Equilibrium (PNE).

**Definition 2.1** (Pure Strategy Nash Equilibrium). A strategy profile $\boldsymbol{b}^* = (\boldsymbol{b}_1^*, \dots, \boldsymbol{b}_I^*)$, with $\boldsymbol{b}_i^* \in \mathbb{R}_{\geq 0}^T$, is a PNE for $\mathcal{G}(\boldsymbol{C})$ if $\boldsymbol{b}_i^*$ is a best response to $\boldsymbol{b}_{-i}^*$ for all $i \in \mathcal{I}$. Specifically, for any alternative strategy $\boldsymbol{b}_i' \in \mathbb{R}_{\geq 0}^T$ satisfying budget constraint (2), $U_i(\boldsymbol{b}_i^*, \boldsymbol{b}_{-i}^*) \geq U_i(\boldsymbol{b}_i', \boldsymbol{b}_{-i}^*)$.

---

[1]In practice, $c_{ti}$ represents strategic platform levers such as **sampling density (Best-of-K)** or **model tiering**. By generating $K$ candidate titles and selecting the one with the maximum predicted CTR, the cost scales linearly with generation attempts while the CTR lift exhibits diminishing marginal returns. This modeling is grounded in industry-standard cost logic (Chen et al., 2023; Nayab et al., 2024). Concavity thus ensures economic tractability while maintaining high industrial fidelity.

## 2.3. The Leader's Optimization Problem

The platform acts as the Stackelberg leader, seeking to maximize net revenue by anticipating the agents' equilibrium response. Let $\mathcal{E}(\boldsymbol{C})$ be the set of PNE bid profiles in the subgame induced by investment trajectory $\boldsymbol{C}$. The platform selects $\boldsymbol{C}$ to maximize total expected payments net of incurred LLM costs:

$$\max_{\boldsymbol{C} \in (\mathbb{R}_{\geq 0})^{I \times T}} \quad \mathcal{J}(\boldsymbol{C}) = \sum_{t \in \mathcal{T}, i \in \mathcal{I}} \mathbb{E}[x_{ti}(\boldsymbol{b}^*) \cdot (e_t^{(2)}(\boldsymbol{b}^*) - c_{ti})] \tag{3}$$

$$\text{s.t.} \quad \boldsymbol{b}^* = \boldsymbol{b}^*(\boldsymbol{C}) \in \mathcal{E}(\boldsymbol{C}) \tag{4}$$

This formulation presents a significant computational challenge: equilibrium constraint (4) implies a non-convex optimization landscape, and mapping $\boldsymbol{C} \mapsto \mathcal{E}(\boldsymbol{C})$ may be set-valued due to equilibrium multiplicity, rendering the objective ill-defined. We address these theoretical and algorithmic challenges subsequently.

# 3. Stackelberg Game Analysis

We analyze the LLM-augmented auction via backward induction. First, we characterize the bidding agents' equilibrium behavior under a fixed platform policy $\boldsymbol{C}$, then address equilibrium selection to define the platform's objective.

In the **Follower's Subgame** (Step 3 of PIM, Section 2.1), the mechanism remains a standard GSP auction with platform-enhanced CTR metrics. Extending theoretical results from (Aggarwal et al., 2019), we establish that for any fixed investment $\boldsymbol{C}$, the subgame admits a Pure Strategy Nash Equilibrium (PNE) with uniform pacing strategies.

**Lemma 3.1** (Existence of Pacing Equilibrium). *For any investment policy $\boldsymbol{C}$, the induced subgame $\mathcal{G}(\boldsymbol{C})$ admits at least one PNE. In equilibrium, every agent $i$ adopts a uniform pacing strategy defined by a dual variable $\lambda_i^*$, bidding $b_{ti}(\lambda_i^*) = v_{ti}/\lambda_i^*$.*

This structural preservation is a key engineering advantage: PIM acts as a modular wrapper around existing autobidders, maintaining essential equilibrium properties (see Appendix D). This enables incremental deployment, mitigating risks associated with full-scale system overhauls.

However, a critical challenge for optimization is that the pacing equilibrium $\boldsymbol{\lambda}^*$ is generally not unique (Conitzer et al., 2022). This multiplicity makes the mapping $\boldsymbol{C} \mapsto \boldsymbol{\lambda}^*(\boldsymbol{C})$ set-valued, rendering gradient estimation ill-defined. Moreover, in concurrent learning, this ambiguity risks instability if the leader optimizes against one equilibrium while agents converge to another. To induce a predictable, unique response, we introduce Tikhonov regularization.

**Definition 3.2** ($\epsilon$-Regularized Dual Objective). For regularization parameter $\epsilon > 0$, the regularized dual objective

for agent $i$ is $\mathcal{D}_{i,\epsilon}(\lambda_i, \boldsymbol{\lambda}_{-i}) = \mathcal{D}_i(\lambda_i, \boldsymbol{\lambda}_{-i}) + \frac{\epsilon}{2}\lambda_i^2$, where $\mathcal{D}_i$ is the standard dual function. Agents minimize this objective, favoring the minimal dual variable among optimal strategies.

We term the game where agents minimize regularized objectives the $\epsilon$-*regularized game*.

**Lemma 3.3** (Unique Regularized Equilibrium). *For $\epsilon > 0$ and fixed $\boldsymbol{C}$, the $\epsilon$-regularized game defined by $\{\mathcal{D}_{i,\epsilon}\}_{i \in \mathcal{I}}$ admits a unique PNE, denoted $\boldsymbol{\lambda}_\epsilon^*(\boldsymbol{C})$.*

*Sketch.* Uniqueness follows from the Diagonal Strict Convexity of the game's pseudogradient (Rosen, 1965) (details in Appendix E.1). □

Finally, we establish that this unique equilibrium serves as a valid approximation of the original game. We define an $\epsilon'$-approximate PNE ($\epsilon'$-PNE) as a profile where no agent improves utility by more than $\epsilon'$ via unilateral deviation. The following theorem provides a stable foundation for learning.

**Theorem 3.4** (Approximation Guarantee). *Let $\boldsymbol{\lambda}_\epsilon^*$ be the unique equilibrium of the $\epsilon$-regularized game. Then $\boldsymbol{\lambda}_\epsilon^*$ constitutes an $\epsilon'$-PNE of the original unregularized game, where $\epsilon' = \frac{\epsilon}{2}\lambda_{\max}^2$, and $\lambda_{\max}$ is a uniform upper bound on dual variables.*

*Sketch.* We bound the regret of playing $\lambda_{i,\epsilon}^*$ in the original game by leveraging the regularized optimality conditions and the bounded strategy space (see Appendix E.2). □

# 4. Online Learning in the Follower's Subgame

We transition to the dynamic online setting where the platform's investment policy and agents' strategies evolve over $T$ rounds. Agents learn optimal bidding in a non-stationary environment with censored feedback (observing $e_t^{(2)}$ only upon winning) and without *a priori* knowledge of future valuations or competitors' bids $\boldsymbol{b}_{-i}$.

To address this, we propose the **Online Dual-Descent Bidding with Regularization (ODDB-R)** algorithm, demonstrating it enables agents to satisfy budget constraints while maximizing value, with rigorous guarantees for global stability and individual regret.

Per Lemma D.1, the optimal bid for a value-maximizing agent $i$ is $b_{ti} = v_{ti}/\lambda_{i,t}$, characterized by a dual variable $\lambda_i \geq 0$. The goal is to generate a sequence $\{\lambda_{i,t}\}_{t=1}^T$ minimizing the cumulative loss of the regularized dual function (Definition 3.2). With budget $B_i$, the instantaneous regularized dual loss $\ell_{i,t}(\lambda_{i,t})$ is:

$$\ell_{i,t}(\lambda_{i,t}) = \lambda_{i,t}\left(\frac{B_i}{T} - x_{ti}(\boldsymbol{b}^t)e_t^{(2)}(\boldsymbol{b}^t)\right) + \frac{\epsilon}{2}\lambda_{i,t}^2, \tag{5}$$

**Algorithm 1** ODDB-R

1: **Input:** Budget $B_i$, horizon $T$, regularization $\epsilon$, max dual bound $\lambda_{\max}$, step size scalar $z$.
2: **Initialize:** $\lambda_{i,1} \in [0, \lambda_{\max}]$ (e.g., $\lambda_{i,1} = 1$).
3: **for** $t = 1$ **to** $T$ **do**
4:     Observe valuation $v_{ti}$ and enhanced CTR $\hat{\alpha}_{ti}$.
5:     Calculate bid: $b_{ti} = v_{ti}/\lambda_{i,t}$.
6:     Submit $b_{ti}$; observe allocation $x_{ti}$ and price $e_t^{(2)}$.
7:     Compute step size: $\eta_t = z/\sqrt{t}$.
8:     Compute stochastic gradient:

$$g_{i,t} = \frac{B_i}{T} - x_{ti}e_t^{(2)} + \epsilon\lambda_{i,t}$$

9:     Update dual variable via projection:

$$\lambda_{i,t+1} = \min\left(\lambda_{\max}, \max\left(0, \lambda_{i,t} - \eta_t g_{i,t}\right)\right)$$

10: **end for**

where $\boldsymbol{b}^t$ is the bid profile. The term $\frac{B_i}{T}$ is the target per-round expenditure, and $x_{ti}(\boldsymbol{b}^t)e_t^{(2)}(\boldsymbol{b}^t)$ is the realized payment. Note $\mathbb{E}[\ell_{i,t}]$ corresponds to the regularized dual function $\mathcal{D}_{i,\epsilon}$ (up to a constant).

The gradient of $\ell_{i,t}$ yields a stochastic estimator for the expected dual function gradient $g_{i,t}$:

$$g_{i,t} = \nabla_{\lambda_{i,t}}\ell_{i,t} = \frac{B_i}{T} - x_{ti}(\boldsymbol{b}^t)e_t^{(2)}(\boldsymbol{b}^t) + \epsilon\lambda_{i,t}. \quad (6)$$

Crucially, payment $x_{ti}(\boldsymbol{b}^t)e_t^{(2)}(\boldsymbol{b}^t)$ is observable under censored feedback: payment is zero if losing ($x_{ti} = 0$), and $e_t^{(2)}$ is revealed if winning ($x_{ti} = 1$).

ODDB-R updates dual variables via Online Gradient Descent (OGD) using a decaying step size $\eta_t = z/\sqrt{t}$ ($z > 0$). To enforce the feasible region $\Omega_i = [0, \lambda_{\max}]$, we perform explicit projection. The update rule is:

$$\lambda_{i,t+1} = \min\left(\lambda_{\max}, \max\left(0, \lambda_{i,t} - \eta_t g_{i,t}\right)\right). \quad (7)$$

Eq. (7) is economically intuitive. If realized payment exceeds the target ($x_{ti}e_t^{(2)} > B_i/T$), the gradient is negative, increasing $\lambda_{i,t+1}$. Higher $\lambda_i$ shades bids ($v_{ti}/\lambda_i$), reducing future spend. Conversely, underspending decreases $\lambda_i$, promoting aggressive bidding. The term $\epsilon\lambda_{i,t}$ acts as a *leak*, pulling the dual variable toward zero for strict convexity and stability.

We analyze ODDB-R's global stability and individual optimality. First, we address system-level convergence. Unlike unregularized auctions where simultaneous updates can cause limit cycles (e.g., Edgeworth cycles), we prove that $\epsilon$-regularization ensures strictly monotone dynamics, guaranteeing convergence to the unique equilibrium.

**Theorem 4.1** (Convergence to $\epsilon$-PNE). *Suppose all agents $i \in \mathcal{I}$ follow ODDB-R with step sizes satisfying $\sum_{t=1}^{\infty} \eta_t = \infty$ and $\sum_{t=1}^{\infty} \eta_t^2 < \infty$. Then, the sequence of joint dual profiles $\boldsymbol{\lambda}_t = (\lambda_{1,t}, \ldots, \lambda_{I,t})$ converges almost surely to the unique Pure Strategy Nash Equilibrium $\boldsymbol{\lambda}_\epsilon^*$ of the $\epsilon$-regularized follower subgame.*

Second, we establish an individual performance guarantee. We show that for any single agent, ODDB-R achieves sublinear regret relative to the best fixed pacing multiplier in hindsight, regardless of competitors' strategies.

**Theorem 4.2** (No-Regret Guarantee). *For any agent $i$, assuming bounded valuations and payments, ODDB-R with step size $\eta_t \propto 1/\sqrt{t}$ achieves $O(\sqrt{T})$ regret against the optimal fixed dual strategy $\lambda_i^*$ satisfying the budget constraint in expectation: $\sum_{t=1}^{T} (U_{i,t}(\lambda_i^*) - U_{i,t}(\lambda_{i,t})) \leq O(\sqrt{T})$, where $U_{i,t}(\lambda) = x_{ti}(\lambda)v_{ti}\hat{\alpha}_{ti}$ is the value obtained at round $t$ given $\lambda$.*

## 5. Online Learning for the Stackelberg Game

We address the bilevel optimization in the online setting. The platform (leader) optimizes its investment policy to maximize a global objective $\mathcal{J}$—net revenue—constrained by agents (followers) minimizing regularized dual functions.

This dynamic Stackelberg game spans a finite horizon $T$. The environment is inherently non-stationary: the platform dynamically adjusts investments $\boldsymbol{c}^t \in \mathbb{R}^I$, while agents concurrently update dual variables $\boldsymbol{\lambda}_t$ via ODDB-R. Key challenges include the leader's lack of analytical access to followers' best-response $\boldsymbol{\lambda}_\epsilon^*(\boldsymbol{C})$ and the failure of single-timescale RL, where frequent updates prevent follower convergence.

To resolve this, we formulate a continuous control problem and propose the **Two-Timescale Stackelberg Learning with Proximal Policy Optimization (TTSL-PPO)** algorithm. We enforce strict timescale separation for stability: followers update strategies at every round $t$ (*fast timescale*), while the leader updates policy parameters only after accumulating sufficient trajectories (*slow timescale*). This ensures followers effectively track the induced equilibrium, providing stable gradient estimates. TTSL-PPO iteratively refines the policy until reaching a Stackelberg Stationary Point (SSP).

### 5.1. Markov Decision Process Formulation

We formulate the leader's problem as a continuous-state, continuous-action MDP with horizon $T$.

**State Space** $\mathcal{S}$. The state $\boldsymbol{s}_t$ at round $t$ captures all platform-observable information: $\boldsymbol{s}_t = (\boldsymbol{\alpha}_t, \boldsymbol{v}_t, \boldsymbol{B}_t, \hat{\boldsymbol{\lambda}}_{t-1})$, comprising baseline CTRs $\boldsymbol{\alpha}_t$, values $\boldsymbol{v}_t$, remaining budgets $\boldsymbol{B}_t$, and inferred duals $\hat{\boldsymbol{\lambda}}_{t-1}$. The platform infers $\hat{\lambda}_{i,t-1} = $

$v_{i,t-1}/b_{i,t-1}$ via Lemma 3.1. Augmenting with $\hat{\lambda}$ ensures Markovianity, as future bids depend on current duals.

**Action Space** $\mathcal{A}$. The leader's action is the investment vector $\boldsymbol{c}^t \in \mathbb{R}^I_{\geq 0}$, with $c_{ti}$ the investment for agent $i$.

**Reward Function.** The immediate reward $r_t$ is the realized net revenue: $r_t = \sum_{i \in \mathcal{I}} x_{ti}(\boldsymbol{b}^t) \left( e_t^{(2)}(\boldsymbol{b}^t) - c_{ti} \right)$, where $x_{ti}$ indicates allocation and $e_t^{(2)}$ is the GSP clearing price.

**Value Function.** The value function $V_{\boldsymbol{\psi}}(\boldsymbol{s})$, parameterized by $\boldsymbol{\psi}$, estimates expected discounted future returns under policy $\pi_{\boldsymbol{\theta}}$, accounting for followers' dynamic response: $V_{\boldsymbol{\psi}}(\boldsymbol{s}_t) \approx \mathbb{E}_{\pi_{\boldsymbol{\theta}},\text{ODDB-R}} \left[ \sum_{j=0}^{T-t} \gamma^j r_{t+j} \mid \boldsymbol{s}_t \right]$, with discount factor $\gamma \in [0, 1)$.

## 5.2. The TTSL-PPO Algorithm

TTSL-PPO (Algorithm 2) couples agents' fast-scale learning with the platform's slow-scale optimization, explicitly acknowledging the subgame's shifting equilibrium.

**Phase 1: Fast-Scale Interaction.** In each round $t$, the platform samples $\boldsymbol{c}_t \sim \pi_{\boldsymbol{\theta}}$. Agents observe enhanced metrics and submit bids. Crucially, agents update duals $\boldsymbol{\lambda}$ at every step via ODDB-R, allowing $\boldsymbol{\lambda}$ to track the equilibrium $\boldsymbol{\lambda}^*(\pi_{\boldsymbol{\theta}})$ induced by the current policy.

**Phase 2: Slow-Scale Optimization.** The platform accumulates experience in buffer $\mathcal{D}$. Policy updates occur every $L$ steps (where $L$ is large). This frequency ensures collected trajectories reflect system behavior near the current policy's equilibrium, filtering transient noise.

**Trajectory-Aware Advantage Estimation.** To stabilize learning, we use Generalized Advantage Estimation (GAE). Let $\delta_t = r_t + \gamma V_{\boldsymbol{\psi}}(\boldsymbol{s}_{t+1}) - V_{\boldsymbol{\psi}}(\boldsymbol{s}_t)$ be the TD error. The advantage is $\hat{A}_t = \sum_{l=0}^{T-t-1} (\gamma\kappa)^l \delta_{t+l}$.

**Proximal Policy Optimization with Joint Objective.** We use a clipped surrogate objective to prevent excessive policy deviation, critical in Stackelberg settings where large jumps can invalidate equilibrium approximations. The joint objective $\mathcal{J}_{PPO}$ is:

$$\max_{\boldsymbol{\theta}, \boldsymbol{\psi}} \mathcal{J}_{PPO}(\boldsymbol{\theta}, \boldsymbol{\psi}) = \mathcal{L}^{CLIP}(\boldsymbol{\theta}) - c_1 \mathcal{L}^{VF}(\boldsymbol{\psi}) + c_2 \mathcal{S}[\pi_{\boldsymbol{\theta}}],$$
(8)

where $\mathcal{L}^{CLIP}$ is the PPO clipped objective, $\mathcal{L}^{VF}$ is squared value loss, and $\mathcal{S}$ is entropy bonus.

## 5.3. Convergence Analysis

We analyze TTSL-PPO convergence to a stationary point, considering asymptotic behavior. Note that our theoretical analysis focuses on the idealized smooth stochastic gradient ascent version, standard in mechanism design literature. The

---

**Algorithm 2** TTSL-PPO
___

1: **Input:** Horizon $T$, Update Interval $L$, follower step sizes $\{\eta_{cnt}\}$, leader learning rate $\beta$, Total Episodes $M$.
2: **Initialize:** Policy $\pi_{\boldsymbol{\theta}}$, Value Function $V_{\boldsymbol{\psi}}$.
3: **Initialize:** Follower duals $\boldsymbol{\lambda} \leftarrow \boldsymbol{1}$ (Maintained continuously).
4: Set global step counter $cnt \leftarrow 0$.
5: Clear trajectory buffer $\mathcal{D}$.
6: **for** episode $m = 1$ to $M$ **do**
7:     **Reset:** Budgets $\boldsymbol{B}_1 \leftarrow \boldsymbol{B}_{initial}$, observe state $\boldsymbol{s}_1$.
8:     **for** round $t = 1$ to $T$ **do**
9:         $cnt \leftarrow cnt + 1$.
10:        Phase 1: Fast-Scale Interaction (Agents adapt)
11:        Sample action $\boldsymbol{c}^t \sim \pi_{\boldsymbol{\theta}}(\cdot|\boldsymbol{s}_t)$.
12:        **Mechanism:** Compute $\hat{\alpha}_{ti}$ given investment $\boldsymbol{c}^t$. Agents bid $b_{ti} = v_{ti}/\lambda_i$.
13:        Auction clears; observe reward $r_t$ and next state $\boldsymbol{s}_{t+1}$.
14:        Agents update $\boldsymbol{\lambda}$ via ODDB-R (Eq. (7)).
15:        Store transition $(\boldsymbol{s}_t, \boldsymbol{c}^t, r_t, \boldsymbol{s}_{t+1})$ in $\mathcal{D}$.
16:        Phase 2: Slow-Scale Optimization (Leader updates)
17:        **if** $|\mathcal{D}| \geq L$ **then**
18:            Compute Generalized Advantages $\hat{A}_l$ for all $l \in \mathcal{D}$.
19:            Update $\boldsymbol{\theta}, \boldsymbol{\psi}$ via Joint PPO Objective (Eq. (8)).
20:            Clear buffer $\mathcal{D}$.
21:        **end if**
22:     **end for**
23: **end for**
24: **Return** Optimized policy $\pi_{\boldsymbol{\theta}}$.

---

practical implementation (Algorithm 2) employs PPO clipping to enhance numerical stability and sample efficiency.

**Definition 5.1** (Stackelberg Stationary Point). A pair $(\pi_{\boldsymbol{\theta}}^*, \boldsymbol{\lambda}^*)$ is a Stackelberg Stationary Point (SSP) if $\boldsymbol{\lambda}^* = \boldsymbol{\lambda}_{\epsilon}^*(\pi_{\boldsymbol{\theta}}^*)$ is the regularized equilibrium response, and $\nabla_{\boldsymbol{\theta}} \mathcal{J}(\pi_{\boldsymbol{\theta}}^*, \boldsymbol{\lambda}^*) = 0$.

**Theorem 5.2** (Convergence of TTSL-PPO). *Assume objective $\mathcal{J}$ and dual functions $\mathcal{D}_{i,\epsilon}$ are smooth and bounded. Let step sizes for followers ($\eta_{cnt}$) and leader ($\beta_k$) satisfy standard Robbins-Monro conditions: $\sum \eta_{cnt} = \infty, \sum \eta_{cnt}^2 < \infty$, and similarly for $\beta_k$. Furthermore, assume timescale separation $\lim_{k \to \infty} \beta_k/\eta_{kL} = 0$. Then, the coupled process $(\boldsymbol{\theta}_k, \boldsymbol{\lambda}_{cnt})$ generated by TTSL-PPO converges almost surely to a Stackelberg Stationary Point (SSP).*

The proof (Appendix G) utilizes the two-timescale stochastic approximation framework, treating the leader's update as a discretization of an ODE driven by the followers' equilibrium response.

# 6. Experimental Evaluation

To rigorously validate our theoretical framework and demonstrate practical efficacy, we conduct a comprehensive empirical analysis. Reported results represent mean values over independent trials, with statistical significance verified via paired $t$-tests ($p < 0.05$). Beyond the core metrics presented here, Appendix B provides extensive supplementary experiments—including standard deviation analysis, investment statistics, and budget utilization rates—to further corroborate robustness. Our experiments address three questions: **Follower Rationality:** Does *ODDB-R* enable agents to converge to a rational bidding equilibrium under budget constraints and platform-induced non-stationarity? **Mechanism Efficacy:** Does *TTSL-PPO* outperform heuristic and myopic strategies in maximizing long-term revenue across varying creative enhancement capabilities? **Architectural Necessity:** Are the specific architectural choices essential for stability and convergence in this Stackelberg game?

## 6.1. Evaluation of Agent Bidding Algorithms

We evaluate agent efficacy within the follower subgame by simulating a standard GSP auction (without platform investments) characterized by $\langle \mathcal{I}, \mathcal{T} \rangle$. The market comprises $I = 10$ autobidding agents competing for a single ad slot over $T = 2,000$ rounds, sufficient to capture both transient learning and steady-state equilibrium.

**Data Generation and Enhancement.** To reflect the heterogeneity of the ad market, we model baseline CTRs, valuations, and budgets as random variables drawn from Normal distributions

**Baselines.** We compare ODDB-R against six prevalent strategies: (1) *Truthful:* Bids true enhanced valuation ($b_{ti} = v_{ti}$). While theoretically optimal in unconstrained auctions, this serves here as an *efficiency upper bound* to benchmark potential welfare losses due to budget constraints. (2) *FixedBid:* Static heuristic bidding a fixed fraction of valuation ($b_{ti} = 0.9v_{ti}$). (3) *Linear Bid:* Bids scaled linearly by remaining budget-to-time ratio. (4) *PID* (Johnson & Moradi, 2005): Control-theoretic approach stabilizing spend rate via a PID controller. (5) *AdWords* (Mehta et al., 2007): Theoretical budget-constrained matching algorithm. (6) *Pacing* (Aggarwal et al., 2019): Multiplicative pacing with PID-adjusted multipliers.

**Evaluation Metrics.** Performance is assessed using: (1) *Total Social Welfare (SW):* Cumulative agent utility. Reported as percentage change vs. *Truthful*; higher values indicate greater efficiency. (2) *Convergence Round (Conv):* The round $t$ after which bid variance falls below $\epsilon = 10^{-2}$. Lower values indicate faster stabilization.

**Experimental Results and Analysis.** Table 1 reveals three behavioral categories: (1) **Static Strategies (Truth-**

*Table 1.* Algorithm Performance Metrics in Agent Subgame.

| Algorithm | SW (vs. Truthful) | Conv |
|---|---|---|
| Truthful | 0.0% | **1** |
| FixedBid | -6.9% | **1** |
| Linear | -9.8% | N/A |
| PID | -30.8% | N/A |
| AdWords | -31.3% | N/A |
| Pacing | -30.1% | 1063 |
| ODDB-R (Ours) | **-29.3%** | **841** |

**ful, FixedBid):** These achieve highest Social Welfare and immediate convergence (Conv=1). However, performance is deceptive; these strategies are fundamentally poor for individual agents as they ignore budget constraints, leading to premature depletion. Thus, they serve merely as theoretical upper bounds. (2) **Non-Converging Dynamic Strategies (Linear, PID, AdWords):** These fail to reach stable equilibrium (N/A). High bidding variance results from over-correction to stochastic valuations, causing instability. (3) **Converging Dynamic Strategies (Pacing, ODDB-R):** Only Pacing and ODDB-R learn stable policies. Crucially, *ODDB-R outperforms Pacing* in efficiency and stability, achieving higher SW (-29.3% vs. -30.1%) and faster convergence (Round 841 vs. 1063). Results confirm that while ODDB-R incurs slight welfare loss compared to idealized static strategies due to learning costs, it effectively guarantees convergence and maximizes individual utility under budget constraints.

## 6.2. Evaluation of Platform Performance

To validate practical efficacy in a realistic setting, we use the **AntM2C** dataset (Huan et al., 2024), a large-scale Alipay dataset containing $\approx$100 million logs. We map items to $I = 10$ competing agents and simulate an environment where creative quality is enhanced via **GPT-4o**. We focus on optimizing *ad titles* for efficiency and real-time feasibility.

**Simulation Setup and Environment Response.** We construct a repeated auction market over $T = 2000$ rounds. For efficient RL training, we constructed a differentiable surrogate function modeling the environment's response. Collecting triplets (Original CTR, Investment Cost, Enhanced CTR) from GPT-4o and a deep CTR model, we fitted a smooth, concave function: $\hat{\phi}(\alpha_{ti}, c_{ti}) = \alpha_{ti} + 0.022 \cdot \alpha_{ti} \cdot (1 - e^{-0.13 \cdot c_{ti}})$. To simulate real-world stochasticity and avoid deterministic overfitting, we added small random perturbations to the function's output during training. This function serves as the ground-truth mechanism, providing algorithms a reliable return estimate, validated to have $< 10\%$ MAPE against actual LLM outputs. See Appendix C for details.

**Baselines and Metrics.** We compare TTSL-PPO against

*Table 2.* Quantitative Performance Comparison by LLM Enhancement Capability (All values are in percentage %).

| Algorithm | Low LLM Capability | | | GPT-4o | | | High LLM Capability | | |
|---|---|---|---|---|---|---|---|---|---|
| | Rev | SW | Click | Rev | SW | Click | Rev | SW | Click |
| GSP No Invest | +0.00 | +0.00 | +0.00 | +0.00 | +0.00 | +0.00 | +0.00 | +0.00 | +0.00 |
| GFP No Invest | -0.32 | -0.04 | -0.02 | -0.32 | -0.04 | -0.02 | -0.32 | -0.04 | -0.02 |
| GSP Uniform Invest | -9.15 | +3.74 | +4.03 | -9.17 | +4.12 | +4.40 | -9.18 | +4.53 | +4.79 |
| GSP Normal Invest | -9.12 | +3.75 | +4.04 | -9.08 | +4.11 | +4.39 | -9.04 | +4.49 | +4.78 |
| GSP Greedy Invest | -7.13 | +1.84 | +1.32 | -7.09 | +2.23 | +1.59 | -7.05 | +2.65 | +1.87 |
| TTSL-PPO (Full) | **+2.62** | **+3.44** | **+3.74** | **+2.66** | **+3.51** | **+3.81** | **+2.70** | +4.20 | +4.44 |
| TTSL-PPO (Single) | +1.81 | +2.73 | +2.94 | +2.14 | +3.44 | +3.66 | +2.51 | +4.22 | +4.46 |
| TTSL-PPO (Partial) | +1.82 | +2.74 | +2.95 | +2.16 | +3.44 | +3.67 | +2.53 | +4.24 | +4.46 |
| TTSL-PPO (No-GAE) | +1.84 | +2.74 | +2.96 | +2.17 | +3.45 | +3.67 | +2.53 | **+4.29** | **+4.53** |

five baselines: (1) *GSP No Invest* (Standard GSP, $c_t = 0$); (2) *GFP No Invest*; (3) *GSP Uniform Invest*; (4) *GSP Normal Invest*; and (5) *GSP Greedy Invest* (myopic profit maximization). The investment range $[0, 0.3]$ was selected based on TTSL-PPO's observed distribution. We report three metrics: *Total Platform Revenue (Rev)*, *Total Social Welfare (SW)*, and *Total Click Volume (Click)*. All metrics are reported as *percentage change* relative to *GSP No Invest*.

**Experimental Results and Analysis.** Table 2 presents results averaged over independent trials. In the GPT-4o setting: (1) **Effectiveness of Creative Enhancement:** Optimizing titles yields significant gains, validating commercial viability: lightweight enhancements unlock economic value without full-content generation latency. (2) **Dominance of TTSL-PPO:** Our algorithm achieves a normalized Revenue increase of **+2.66%**, significantly outperforming all baselines, confirming that strategically endogenizing creative costs creates substantial value. (3) **Ecosystem Expansion:** TTSL-PPO acts as a *win-win* mechanism. Beyond revenue, it increases Social Welfare by **3.51%** and Click Volume by **3.81%**. By selectively enhancing creatives with the highest marginal CTR gain ($\hat{\alpha}_{ti}$), the platform expands total surplus. (4) **The Pitfall of Myopic Optimization:** Notably, *GSP Greedy Invest* causes a revenue *loss* of -7.09%. This result underscores the necessity of analyzing the Stackelberg equilibrium. Myopic strategies maximizing immediate surplus ignore agents' strategic responses. Aggressive extraction forces agents into a subgame equilibrium with severe bid shading, collapsing clearing prices. Only TTSL-PPO balances investment cost with sustainable bidding incentives.

### 6.3. Scalability and Market Heterogeneity

To evaluate the robustness of TTSL-PPO in more complex and high-dimensional environments, we extend our simulation to a **3-slot GSP auction** with $I = 100$ **heterogeneous agents**. These agents differ in their budget sizes, value distributions, and responsiveness to creative enhancements. This

setting aligns with realistic search engine conditions where multiple slots are available and the bidder pool is larger.

*Table 3.* Scalability Test Results.

| Algorithm | Rev | SW | Click |
|---|---|---|---|
| GSP No Invest | +0.00% | +0.00% | +0.00% |
| GFP No Invest | -0.53% | -0.04% | -0.02% |
| GSP Uniform Invest | -11.20% | +4.08% | +4.38% |
| GSP Normal Invest | -11.08% | +4.11% | +4.39% |
| GSP Greedy Invest | -9.87% | +2.24% | +1.59% |
| **TTSL-PPO (Ours)** | **+1.07%** | **+3.52%** | **+3.81%** |
| TTSL-PPO (Single) | +0.92% | +3.44% | +3.67% |
| TTSL-PPO (Partial) | +0.93% | +3.43% | +3.68% |
| TTSL-PPO (No-GAE) | +0.92% | +3.44% | +3.67% |

As shown in Table 3, even with 100 heterogeneous agents and multiple slots, TTSL-PPO maintains its superiority over non-strategic baselines. Although the relative revenue gain (+1.07%) is lower than in the 10-agent case due to increased competition and market density, the qualitative conclusions remain consistent. This demonstrates that our two-timescale Stackelberg learning framework effectively scales to larger bidder populations.

### 6.4. End-to-End Validation and Surrogate Fidelity

While the surrogate function $\hat{\phi}$ ensures training efficiency, it is crucial to verify that its predictions hold in a truly generative environment. We conducted a **fully end-to-end experiment** where we bypassed the surrogate and directly invoked the **GPT-4o API** in real-time to generate enhanced titles for each winning bidder. These raw generated titles were then processed by the DeepFM model to determine the actual realized CTR.

The results in Table 4 not only validate the effectiveness of TTSL-PPO (+2.04% revenue gain) in a more realistic setting

_Table 4._ End-to-End Performance Comparison.

| Algorithm | Rev | SW | Click |
|---|---|---|---|
| GSP No Invest | +0.00% | +0.00% | +0.00% |
| GFP No Invest | -0.32% | -0.04% | -0.02% |
| GSP Uniform Invest | -11.12% | +3.74% | +4.03% |
| GSP Normal Invest | -11.06% | +3.75% | +4.04% |
| GSP Greedy Invest | -9.92% | +1.84% | +1.32% |
| **TTSL-PPO (Ours)** | **+2.04%** | **+3.43%** | **+3.73%** |
| TTSL-PPO (Single) | +1.66% | +2.76% | +2.98% |
| TTSL-PPO (Partial) | +1.67% | +2.77% | +3.00% |
| TTSL-PPO (No-GAE) | +1.64% | +2.72% | +2.95% |

but also confirm the reliability of our surrogate function. The surrogate captures the essential commercial trends of LLM-based creative enhancement, serving as a cost-effective and reproducible initialization for training in real-world ad systems.

### 6.5. Ablation Study

To validate framework components, we compare the full algorithm against three variants. Results are reported in the _GPT-4o_ column of Table 2.

**Ablation Algorithms.** (1) **TTSL-PPO (Single):** Single-timescale variant updating policy $\pi_\theta$ at every step $t$ concurrent with agent updates, removing timescale separation. (2) **TTSL-PPO (Partial):** Variant restricting state observation to $(\boldsymbol{\alpha}_t, \boldsymbol{v}_t, \boldsymbol{B}_t)$, removing inferred dual variables $\hat{\boldsymbol{\lambda}}_{t-1}$ to test the importance of observing budget constraints. (3) **TTSL-PPO (No-GAE):** Replaces GAE with standard Monte-Carlo returns to test variance reduction.

**Experimental Results and Analysis.** Table 2 shows **TTSL-PPO** consistently outperforms variants in Revenue (+2.66%). (1) **Necessity of Timescale Separation: TTSL-PPO (Single)** performs worst (Revenue +2.14%). Updating the policy at every step creates a highly non-stationary environment, preventing agent convergence and leading to erratic bidding. This confirms the two-timescale design is critical. (2) **Importance of State Augmentation: TTSL-PPO (Partial)** shows degraded performance (Revenue +2.16%). Without inferred dual variables $\hat{\boldsymbol{\lambda}}_{t-1}$, the platform cannot distinguish high-valuation budget-constrained agents, failing to target investments effectively. (3) **Role of Variance Reduction: TTSL-PPO (No-GAE)** yields lower Revenue (+2.17%), though higher SW (+3.45%) and Clicks (+3.67%). Without GAE, policy optimization is less precise, tending to over-invest. This increases welfare but fails to convert it into revenue efficiently. TTSL-PPO with GAE better aligns investment with net revenue maximization, avoiding _wasteful_ welfare generation.

### 6.6. Impact of Generative AI Performance Variability

We assess robustness across LLM capabilities by extending experiments beyond GPT-4o. We simulated _weaker_ and _stronger_ LLMs by modifying enhancement function parameters $\hat{\phi}(\alpha_{ti}, c_{ti}) = \alpha_{ti} + \beta \cdot \alpha_{ti} \cdot (1 - e^{-\gamma \cdot c_{ti}})$. We set $\beta = 0.012, \gamma = 0.12$ for **Low LLM Capability** and $\beta = 0.032, \gamma = 0.14$ for **High LLM Capability** (vs. $\beta = 0.022, \gamma = 0.13$ for GPT-4o).

**Experimental Results and Analysis.** Table 2 summarizes performance across capability levels. (1) **Monotonic Performance Scaling:** Stronger generative models act as force multipliers. Shifting from _Low_ to _High_ capability monotonically increases performance gains for all algorithms. Powerful LLMs yield higher CTR lift for the same cost, driving competition. (2) **Algorithmic Adaptability:** Crucially, TTSL-PPO maintains dominance across all settings. Even in the _Low Capability_ scenario, it achieves normalized Revenue of +2.62%, significantly outperforming baselines. This proves the framework correctly identifies marginal productivity to allocate costs only where revenue lift exceeds expense. (3) **Optimization Objective Alignment:** While **TTSL-PPO (No-GAE)** sometimes yields higher SW or Clicks, full **TTSL-PPO** consistently achieves the highest Revenue. This highlights a key distinction: maximizing welfare often requires over-investment that diminishes net profit. TTSL-PPO effectively curbs inefficient investments, strictly optimizing the platform's net revenue objective $\mathcal{J}$.

## 7. Discussion and Conclusion

This paper establishes a rigorous game-theoretic framework for integrating LLM-powered creative enhancement into ad auctions. We proposed the Platform-Investment Mechanism (PIM) to strategically manage the high costs of generative AI, and developed the TTSL-PPO algorithm to optimize platform investment while accounting for the equilibrium response of autobidding agents. Our extensive empirical validation confirms that this approach aligns incentives, boosting platform revenue, advertiser welfare, and user clicks.

While our analysis focused on the GSP auction, the proposed framework exhibits broad applicability. The core methodology—specifically the two-timescale learning architecture—is agnostic to the underlying auction rules. It can be readily extended to VCG auctions (Aggarwal et al., 2019), GFP auctions (Alimohammadi et al., 2023), or multi-slot allocation settings by simply substituting the auction mechanism (Line 13 in Algorithm 2) with the appropriate alternative. This flexibility positions our work as a foundational step toward designing economically robust, next-generation ad systems powered by generative AI.

## Acknowledgements

This work was supported by National Natural Science Foundation of China (No.62472428), Public Computing Cloud, Renmin University of China, the fund for building world-class universities (disciplines) of Renmin University of China.

## Impact Statement

This paper presents work whose goal is to advance the field of Machine Learning by introducing a novel mechanism for LLM-enhanced creative generation in ad auctions. Our work contributes to the development of efficient and sustainable AI-driven economic systems by endogenizing the computational costs of generative models. While there are many potential societal consequences of our work, such as improved user experience through higher-quality ad content and enhanced platform efficiency, none which we feel must be specifically highlighted here.

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

# A. Implementation and Architectural Details

This appendix provides a comprehensive specification of the experimental framework, algorithm architectures, and training protocols used to generate the results in Section 6. We detail the hyperparameters and implementation choices to ensure full reproducibility and provide deeper context for the experimental findings.

## A.1. Experimental Setup Details

### A.1.1. SIMULATION ENVIRONMENT

We simulate a repeated Generalized Second-Price (GSP) auction market characterized by the tuple $\langle \mathcal{I}, \mathcal{T} \rangle$. The market consists of $I = 10$ autobidding agents competing for a single ad slot over a finite campaign horizon of $T = 2,000$ auction rounds. This horizon was selected to be sufficiently long to observe both the transient learning phase of the agents and the steady-state equilibrium behavior of the system. All reported experimental results represent the mean values calculated over **independent repeated trials** with different random seeds to ensure statistical reliability and robustness.

### A.1.2. DATA GENERATION

We employ two distinct data generation protocols tailored to the specific objectives of the agent-level and platform-level experiments.

**For Agent Algorithm Evaluation:** To rigorously stress-test the bidding algorithms under controlled stochasticity, we generate synthetic data where baseline metrics follow truncated Normal distributions. Specifically, for each agent $i$ at round $t$:

- Baseline Click-Through Rate: $\alpha_{ti} \sim \mathcal{N}(0, 1)$, clipped to the interval $[0, 1]$.

- Value-Per-Click: $v_{ti} \sim \mathcal{N}(0, 1)$, clipped to the interval $[0, 5]$.

- Total Budget: $B_i \sim \mathcal{N}(0, 1)$, clipped to the interval $[100, 300]$.

This setup ensures diverse market conditions, ranging from high-value/tight-budget scenarios to low-value/loose-budget scenarios.

**For Platform Algorithm Evaluation:** To validate the platform's investment policy in a realistic industrial setting, we utilize real-world user interaction logs from the **AntM2C** dataset (Huan et al., 2024). The detailed preprocessing pipeline, feature engineering, and ground-truth CTR simulation for this dataset are provided in Appendix C.

### A.1.3. BASELINES

We compare our proposed methods against a comprehensive set of baselines for both agent strategies and platform policies.

**Agent Baselines:**

- **Truthful Bidding:** The agent bids their true enhanced valuation, $b_{ti} = v_{ti}$. While this serves as a **Theoretical Efficiency Upper Bound** for unconstrained auctions, it fails to account for budget constraints and is typically infeasible in practice. Comparing against it highlights the welfare loss due to hard budget constraints.

- **Fixed Budget Bid:** A static heuristic where the bid is a fixed fraction of the valuation, $b_{ti} = v_{ti} \times \gamma$, where $\gamma$ is a tuned constant (set to 0.9 in our experiments). This strategy lacks adaptivity to dynamic competition or valuation shifts.

- **Linear Bid:** A time-aware pacing strategy where bids are scaled linearly by the remaining budget relative to the remaining time. The bid is calculated as $b_{ti} = v_{ti} \times \min\left(\alpha \frac{B_{i,t}}{T-t}, 2.0\right)$. This is a standard industry baseline for smooth delivery.

- **PID Bid** (Johnson & Moradi, 2005)**:** A control-theoretic approach that uses a Proportional-Integral-Derivative controller to minimize the error between the current spend rate and a target uniform spend rate. The bid adjustment is derived dynamically from the PID error signal $e_t$.

- **Pacing (Aggarwal et al., 2019):** A multiplicative pacing strategy where the bid is $b_{ti} = v_{ti} \times \xi_t$. The pacing multiplier $\xi_t$ is adjusted via a PID controller to track the budget schedule. This is a common implementation in modern autobidders.

- **MSVV (Mehta et al., 2007):** The standard algorithm for the budget-constrained AdWords problem. It scales bids using the function $\psi(z) = 1 - e^{z-1}$, where $z$ is the fraction of the budget spent so far. This represents the theoretical state-of-the-art for online matching in static settings.

**Platform Baselines:**

- **GSP No Invest:** The platform makes zero investment ($c_t = 0$). This represents the standard GSP auction and serves as the performance floor.

- **GFP No Invest:** Zero investment under a Generalized First-Price (GFP) auction. Comparing against this benchmarks the inherent revenue properties of the auction mechanism itself.

- **GSP Uniform Invest:** A stateless random strategy where investments are sampled uniformly: $c_{ti} \sim U(0, 0.3)$.

- **GSP Normal Invest:** A concentrated random strategy where investments are sampled from $\mathcal{N}(0, 1)$ and clipped to $[0, 0.3]$.

- **GSP Greedy Invest:** A myopic optimization strategy that selects $c_t$ to maximize the immediate net profit: $c_t = \arg\max_c(\hat{\alpha}_{ti} v_{ti} - c_t)$. This strategy optimizes for immediate returns without considering the dynamic equilibrium effects of the game.

**Ablation Baselines:**

- **TTSL-PPO (Single):** A variant that updates the policy $\pi_\theta$ at every step $t$, concurrent with agent updates, removing the timescale separation.

- **TTSL-PPO (Partial):** A variant where the state input removes the inferred dual variables $\hat{\boldsymbol{\lambda}}_{t-1}$, testing the value of observing agent budget constraints.

- **TTSL-PPO (No-GAE):** A variant that replaces Generalized Advantage Estimation (GAE) with standard Monte-Carlo returns, testing the impact of variance reduction.

A.1.4. EVALUATION METRICS

We employ a multi-dimensional metric system to assess performance from the perspectives of the platform, the agents, and the overall ecosystem:

- **Total Platform Revenue (Rev):** The cumulative net revenue $\sum_{t=1}^{T} r_t = \sum_{t=1}^{T} \sum_i x_{ti}(e_t^{(2)} - c_{ti})$, which is the Leader's primary optimization objective $\mathcal{J}$.

- **Total Social Welfare (SW):** The sum of cumulative utility for all agents, representing the aggregate welfare of the advertisers.

- **Total Click Volume (Click):** The aggregate number of clicks generated, serving as a proxy for user engagement and ecosystem vitality.

- **Convergence Round (Conv):** The round $t$ after which the variance of the agents' bids $\boldsymbol{b}^t$ falls below a threshold $\epsilon = 10^{-2}$, indicating system stability.

## A.2. Agent Implementation: ODDB-R

The advertiser agents implement the *Online Dual-Descent Bidding with Regularization* (ODDB-R) algorithm. To strictly align with the theoretical convergence guarantees, the implementation follows these specifications:

**Dual Variable Dynamics.** Each agent $i$ maintains a scalar dual variable $\lambda_{i,t}$.

- **Initialization:** $\lambda_{i,1}$ is initialized to 1.0 for all agents, representing an unbiased initial valuation.

- **Update Rule:** At the end of each round $t$, agents update their dual variables using the projected gradient descent rule defined in **Eq.** (7).

- **Step Size Schedule:** We use the time-decaying step size $\eta_t = \eta_0 / \sqrt{t}$, where $\eta_0 = 0.1$. This satisfies the square-summable but not summable condition required for convergence.

- **Constraints:** The dual variable is strictly clipped to the range $[0, \lambda_{\max}]$ with $\lambda_{\max} = 10.0$. The regularization parameter is set to $\epsilon = 0.1$.

## A.3. Platform Algorithm: TTSL-PPO

The platform's policy is trained using the *Two-Timescale Stackelberg Learning with Proximal Policy Optimization* (TTSL-PPO) algorithm.

**Network Architectures.** We use an Actor-Critic architecture with separate parameters for the policy ($\pi_{\boldsymbol{\theta}}$) and value function ($V_{\boldsymbol{\psi}}$).

- **State Input:** The input state $\boldsymbol{s}_t$ has dimension $4I$ ($I = 10 \Rightarrow 40$ dimensions), comprising normalized budgets $\boldsymbol{B}_t$, baseline CTRs $\boldsymbol{\alpha}_t$, baseline values $\boldsymbol{v}_t$, and inferred agent duals $\hat{\boldsymbol{\lambda}}_{t-1}$.

- **Hidden Layers:** Both networks employ two fully connected hidden layers with 64 units each. We use `ReLU` activations followed by `LayerNorm` to stabilize training dynamics.

- **Actor Output:** The policy network outputs a mean vector $\boldsymbol{\mu}_t \in \mathbb{R}^I$. A `Sigmoid` activation scales these to $[0, 1]$, multiplied by $c_{\max}$ to obtain investments $\boldsymbol{c}_t$. The log-standard deviation is a learnable parameter initialized to 0.0.

- **Critic Output:** A single linear unit estimating the discounted future revenue.

**Optimization Mechanics.**

- **Learning Rates:** We use separate learning rates to stabilize training: $1 \times 10^{-4}$ for the Actor and $3 \times 10^{-4}$ for the Critic. A lower actor learning rate prevents drastic policy shifts that could destabilize the agents' learning process.

- **PPO Updates:** Updates occur every $L$ rounds. During an update, we perform $K = 10$ optimization epochs (passes) over the collected batch using Minibatch Stochastic Gradient Descent.

- **Advantage Estimation:** We use Generalized Advantage Estimation (GAE) with $\gamma = 0.99$ and $\kappa = 0.95$ to balance bias and variance in the gradient estimates.

## A.4. Training Protocol and Convergence

The training follows the two-timescale interaction loop:

1. **Fast Timescale (Agents):** Agents update $\boldsymbol{\lambda}_t$ at every step $t = 1 \ldots T$.

2. **Slow Timescale (Platform):** The platform accumulates experience and updates its policy $\boldsymbol{\theta}$ only after every $L = 50$ rounds. This interval $L$ allows the agents' dual variables to adjust to the current policy, providing a meaningful gradient for the leader.

**Convergence Criterion.** Training is terminated when the policy stabilizes, defined as: (1) the rolling average of total revenue changes by less than 1% over 10 episodes, and (2) the magnitude of policy updates $\|\boldsymbol{\theta}_{k+1} - \boldsymbol{\theta}_k\|_2$ falls below $10^{-3}$.

## A.5. Hyperparameter Settings

Table 5 summarizes the hyperparameters used in our experiments. We provide definitions for key parameters to clarify their role in the learning process.

*Table 5.* Hyperparameters for TTSL-PPO Algorithm, ODDB-R Agents, and Simulation Environment.

| Parameter | Value | Description |
|---|---|---|
| *TTSL-PPO Optimization (Leader)* | | |
| Actor Learning Rate | $1 \times 10^{-4}$ | Step size for policy network optimization. |
| Critic Learning Rate | $3 \times 10^{-4}$ | Step size for value function network optimization. |
| Discount Factor ($\gamma$) | 0.99 | Weighting factor for future rewards in return calculation. |
| GAE Parameter ($\kappa$) | 0.95 | Controls bias-variance trade-off in advantage calculation. |
| PPO Clip Ratio | 0.2 | Limits the magnitude of policy updates to ensure stability. |
| Entropy Coefficient ($c_2$) | 0.01 | Weight of entropy bonus to encourage exploration. |
| Value Loss Coefficient ($c_1$) | 0.5 | Weight of value function error in the total loss. |
| Minibatch Size | 64 | Number of samples used for each gradient update. |
| Update Interval ($L$) | 50 rounds | Frequency of leader policy updates (slow timescale). |
| *ODDB-R Agents (Followers)* | | |
| Initial Dual Variable ($\lambda_{i,1}$) | 1.0 | Starting value for the Lagrange multiplier. |
| Step Size Scaling ($\eta_0$) | 0.1 | Base learning rate for dual variable updates. |
| Regularization ($\epsilon$) | 0.1 | Coefficient for the Tikhonov regularization term. |
| Max Dual Bound ($\lambda_{\max}$) | 10.0 | Upper limit for clipping dual variables. |
| *Simulation Environment* | | |
| Number of Agents ($I$) | 10 | Number of competing advertisers in each auction. |
| Auction Horizon ($T$) | 2000 | Total number of rounds per campaign simulation. |
| Total Training Episodes | 2000 | Number of full campaign simulations for training. |

# B. Additional Experiments

This appendix provides supplementary experimental results to further validate the stability, robustness, and internal dynamics of our proposed framework.

## B.1. Convergence Dynamics of Agent Bidding

To rigorously analyze the stability of the follower subgame and validate the theoretical claims in Theorem 4.1, we visualize the learning trajectory of the ODDB-R algorithm. Figure 2 illustrates the evolution of the dual variables $\lambda_{i,t}$ for 10 representative agents over the entire auction horizon of $T = 2,000$ rounds.

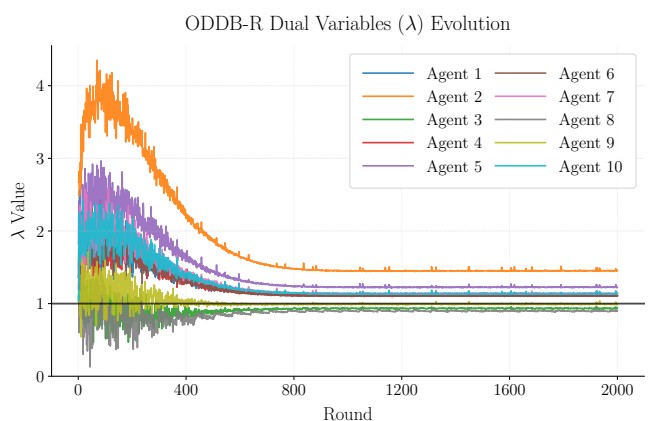

*Figure 2.* Convergence of Dual Variables ($\lambda_{i,t}$) for 10 agents using ODDB-R. Different colors represent different agents.

The dynamics exhibit a clear two-phase convergence process:

*Table 6.* Standard Deviation of the Rev.

| Strategy | Low LLM Capability | GPT-4o | High LLM Capability |
|---|---|---|---|
| GSP No Invest | 0.10443 | 0.10443 | 0.10443 |
| GFP No Invest | 0.11191 | 0.11191 | 0.11191 |
| GSP Uniform Invest | 0.11176 | 0.11205 | 0.11233 |
| GSP Normal Invest | 0.11009 | 0.11042 | 0.11071 |
| GSP Greedy Invest | 0.17350 | 0.17378 | 0.17408 |
| TTSL-PPO (Full) | **0.10456** | **0.10509** | **0.10555** |
| TTSL-PPO (Single) | 0.10469 | 0.10515 | 0.10567 |
| TTSL-PPO (Partial) | 0.10463 | 0.10499 | 0.10561 |
| TTSL-PPO (No-GAE) | 0.10469 | 0.10522 | 0.10574 |

1. **Exploration Phase (Rounds 0-800):** Initially, the dual variables $\lambda_{i,t}$ fluctuate significantly. This volatility arises as agents explore the competitive landscape, attempting to calibrate their bids against unknown competitor strategies and the platform's dynamic investment policy. The regularization term in the ODDB-R update rule is critical here, preventing unbounded oscillations during this high-variance period.

2. **Stabilization Phase (Rounds 800-2000):** The dual variables stabilize into distinct, non-degenerate strata. The values of $\lambda_{i,t}$ converge to the optimal shadow prices associated with each agent's budget constraint. Agents with tight budget-to-value ratios converge to high $\lambda$ values, indicating that aggressive bid shading is necessary to conserve budget over the horizon. Conversely, "wealthy" agents with ample budgets relative to their potential value converge to $\lambda$ values near 1.0, enabling them to bid close to their true enhanced valuation.

This empirical evidence of stable convergence confirms that ODDB-R successfully steers the multi-agent system to a unique Pure Strategy Nash Equilibrium (PNE).

### B.2. Stability Analysis: Revenue Standard Deviation

To assess the stability of platform revenue under different strategies, we recorded the standard deviation of the Total Platform Revenue across independent trials. Low variance indicates a robust policy that performs consistently despite the stochasticity of user arrivals and agent valuations. Table 6 presents the results across three levels of LLM enhancement capability (Low, Medium/GPT-4o, High).

**Results Analysis.**

1. **High Stability of TTSL-PPO:** Crucially, the standard deviation of our proposed **TTSL-PPO (Full)** (e.g., 0.10509 under Medium Lift) is almost identical to that of the passive baseline *GSP No Invest* (0.10443). This demonstrates that our algorithm introduces negligible additional volatility into the auction system while significantly boosting revenue. The timescale separation effectively isolates the platform from the high-frequency noise of agent learning.

2. **Instability of Greedy Strategies:** In sharp contrast, *GSP Greedy Invest* exhibits a drastically higher standard deviation ($\approx 0.174$), nearly 70% higher than TTSL-PPO. This confirms that myopic optimization destabilizes the market, leading to erratic revenue outcomes as agents aggressively adjust bids to counter the platform's unpredictable extraction behavior.

3. **Impact of Ablations:** The ablated variants (Single, Partial, No-GAE) show slightly higher variance than the full model. This further validates that the complete architectural design—specifically the two-timescale separation and GAE—contributes not only to higher performance but also to greater system stability.

## C. Details of Experimental Evaluation on GPT-4o

This appendix provides a detailed specification of the experimental pipeline used to construct the simulation environment based on the **AntM2C** dataset. We describe the data processing, the training of the Ground Truth Simulator, the specific LLM integration, and the auction configuration.

### C.1. Dataset Processing and Feature Engineering

We utilize the **AntM2C** dataset (Huan et al., 2024), which contains user exposure-click logs from Alipay's multi-scenario ecosystem. To create a computationally feasible yet statistically representative environment, we performed the following steps:

1. **Data Filtering and Splitting:** We sampled a subset of $10^7$ records. The data was sorted chronologically by `log_time`. We utilized the first 80% (approx. 8 million samples) to train the CTR prediction model and the subsequent 20% (approx. 2 million samples) for the online auction simulation.

2. **Feature Engineering:** We extracted two categories of features to model user preference and ad quality:
   - **ID Features (Categorical):** We utilized 37 anonymized features, including user behavior sequences (e.g., `bill_entity_seq`), user profiles, and advertiser attributes. The `scene` ID (0-4) is used as a global context feature.
   - **Semantic Features (Textual):** To enable LLM-based enhancement, we extracted the raw text of the `item_title` (ad copy), `item_entity_names` (product entities), and the user's search history (`query_entity_seq`).

### C.2. CTR Prediction Model: Architecture and Training

We trained a deep neural network to serve as the "Ground Truth Simulator," predicting the click-through rate (CTR) $\hat{\alpha}_{ti}$ for any given ad title (original or enhanced).

**Model Architecture.** We employed a DeepFM (Deep Factorization Machine) architecture extended with a semantic encoder:

- **FM Component:** Captures second-order feature interactions explicitly.
- **Deep Component:** A Multi-Layer Perceptron (MLP) with dimensions $[1024, 512, 256]$ and ReLU activations to capture high-order interactions.

Crucially, to incorporate semantic information, we extended the standard DeepFM:

- **Semantic Encoder:** We used a pre-trained `BERT-base-chinese` model to encode the `item_title`. The `[CLS]` token output (dimension 768) is projected down to the same dimension as the ID embeddings (dim=32) via a linear layer.
- **Feature Fusion:** The projected semantic embedding is concatenated with the 27 ID embeddings and the scene embedding before being fed into the FM and Deep layers.

**Training Protocol.** The model was trained on the 8-million-sample split using the Adam optimizer with a learning rate of $10^{-4}$ and a batch size of 4096. We minimized the Binary Cross-Entropy (Logloss). The trained model achieved an **AUC of 0.745** on the test set, ensuring a realistic response to creative changes.

### C.3. LLM-Enhanced Creative Generation

We utilized **GPT-4o** as the generative foundation model to perform the creative enhancement task (rewriting ad titles).

**Prompt Engineering.** We designed a structured prompt to guide the LLM in optimizing the ad title based on the specific user context. The prompt template is defined as follows:

> *"Role: Expert Ad Copywriter.*
> *Task: Rewrite the ad title to increase CTR for the following context.*
> *Context:*
> *- **Product Entities**: [`item_entity_names`]*
> *- **User Interest**: The user recently searched for [`query_entity_seq`] and used services [`service_entity_seq`].*

*- **Original Title**: [item_title]*
*Constraint: Keep it under 50 characters. Highlight benefits relevant to the user's history.*
*Output Format: Return ONLY ONE rewritten title. Just output the title on a single line. No numbering/bullets."*

**Cost Quantification.** The investment cost $c_{ti}$ represents the actual monetary cost incurred by the platform for the enhancement service. We quantified this based on the official API pricing of GPT-4o for the tokens consumed by the prompt and the generated title.

**Function Fitting and Validation.** To facilitate large-scale RL training, which requires millions of interactions, we distilled the LLM's capabilities into a differentiable surrogate function $\hat{\phi}$. This is a standard practice to overcome the latency and cost bottlenecks of real-time API calls during training. The pipeline proceeded as follows:

1. **Generate:** We sampled items and user contexts, using GPT-4o to generate enhanced titles.

2. **Predict:** We passed both original and enhanced titles to the trained DeepFM to obtain $\alpha_{ti}$ and $\hat{\alpha}_{ti}$.

3. **Fit:** We collected triplets of $(\alpha_{ti}, c_{ti}, \hat{\alpha}_{ti})$ and fitted the curve $\hat{\phi}$ using non-linear least squares.

4. **Validation Experiment:** To address concerns regarding the realism of using a fitted function, we conducted a rigorous validation on a hold-out test set of 100 randomly sampled auctions. For these auctions, we executed the actual GPT-4o inference and compared the resulting CTR gains against the predictions of our surrogate function $\hat{\phi}$. The analysis confirmed a strong alignment, with a Mean Absolute Percentage Error (MAPE) of $< 10\%$, validating that our simulation faithfully reflects the dynamics of authentic generative AI enhancement.

### C.4. Auction Environment Configuration

The simulation environment was configured as follows:

- **Simulation Horizon:** The test data is grouped into $T = 2,000$ auction rounds.

- **Agent Selection:** In each round, we randomly sampled 10 active item_ids from the same scene to participate as competing advertisers ($I = 10$).

- **Budgets and Valuations:** To ensure budget constraints are strictly binding, we generate agent valuations $v_{ti}$ and budgets $B_i$ using truncated normal distributions. Specifically, $v_{ti}$ is drawn from the range $[5, 10]$ and $B_i$ from $[100, 300]$. Baseline click-through rates (CTRs) are then obtained by querying the pre-trained DeepFM model with sampled user context and item features. The baseline CTRs used in the TTSL-PPO algorithm are generated by the trained DeepFM model, based on real-world user features and item-specific attributes.

## D. The Follower's Subgame: Structural Analysis

In this appendix, we provide the detailed structural analysis of the follower's subgame $\mathcal{G}(C)$ introduced in Section 3. We assume the platform's investment policy $C$ is fixed, resulting in deterministic enhanced metrics $\hat{\alpha}_{ti}$ for all agents $i \in \mathcal{I}$ and rounds $t \in \mathcal{T}$ (recall that valuations $v_{ti}$ remain exogenous).

We proceed by first deriving the optimal response of a single agent to the aggregate market competition, transforming the strategy space from high-dimensional bids to scalar dual variables. We then analyze the resulting game in the dual space to establish existence guarantees.

### D.1. Optimal Bidding Strategy

Each agent $i$ aims to maximize their total value subject to a hard budget constraint $B_i$. Let $\boldsymbol{b}_{-i}$ denote the bid profiles of all competing agents. The primal optimization problem for agent $i$ is:

$$\max_{\{b_{ti} \geq 0\}_{t \in \mathcal{T}}} \quad \sum_{t \in \mathcal{T}} \mathbb{E}_{\boldsymbol{b}_{-i}} \left[ x_{ti}(\boldsymbol{b}^t) v_{ti} \hat{\alpha}_{ti} \right]$$

$$\text{s.t.} \quad \sum_{t \in \mathcal{T}} \mathbb{E}_{\boldsymbol{b}_{-i}} \left[ x_{ti}(\boldsymbol{b}^t) e_t^{(2)}(\boldsymbol{b}^t) \right] \leq B_i,$$

where $x_{ti}(\boldsymbol{b}^t) \in \{0, 1\}$ is the allocation rule and $e_t^{(2)}(\boldsymbol{b}^t)$ is the second-price payment rule of the GSP auction. Note that the value derived from an impression is the product of the per-click value $v_{ti}$ and the enhanced CTR $\hat{\alpha}_{ti}$.

We address this constrained optimization problem using the method of Lagrange multipliers. Let $\lambda_i \geq 0$ be the dual variable (shadow price) associated with agent $i$'s budget constraint. The Lagrangian function $\mathcal{L}_i$ is given by:

$$\mathcal{L}_i(\boldsymbol{b}_i, \lambda_i; \boldsymbol{b}_{-i}) = \sum_{t \in \mathcal{T}} \mathbb{E}\left[x_{ti}(\boldsymbol{b}^t)v_{ti}\hat{\alpha}_{ti}\right] - \lambda_i \left(\sum_{t \in \mathcal{T}} \mathbb{E}\left[x_{ti}(\boldsymbol{b}^t)e_t^{(2)}(\boldsymbol{b}^t)\right] - B_i\right).$$

Rearranging terms to group them by auction round $t$, we observe that maximizing the Lagrangian is equivalent to maximizing a pointwise surplus function plus a constant budget term:

$$\mathcal{L}_i(\boldsymbol{b}_i, \lambda_i; \boldsymbol{b}_{-i}) = \sum_{t \in \mathcal{T}} \mathbb{E}\left[x_{ti}(\boldsymbol{b}^t)\left(v_{ti}\hat{\alpha}_{ti} - \lambda_i e_t^{(2)}(\boldsymbol{b}^t)\right)\right] + \lambda_i B_i. \tag{9}$$

The term $\left(v_{ti}\hat{\alpha}_{ti} - \lambda_i e_t^{(2)}(\boldsymbol{b}^t)\right)$ represents the net Lagrangian utility of winning item $t$. The following lemma characterizes the optimal bidding strategy that maximizes this per-round utility.

**Lemma D.1** (Optimal Uniform Pacing). *For a value-maximizing agent $i$ participating in a GSP auction with a fixed dual variable $\lambda_i \geq 0$, the optimal per-click bid $b_{ti}$ at round $t$ is given by:*

$$b_{ti}(\lambda_i) = \frac{v_{ti}}{\lambda_i}.$$

*Proof.* By linearity of expectation, maximizing Eq. (9) decouples into maximizing the term inside the summation for each auction $t$ independently, conditional on the realization of competitors' bids. Let $e_t^{(2)}$ denote the clearing price (the highest score among competitors) for a specific realization. Agent $i$ wins ($x_{ti} = 1$) if their rank score $e_{ti} = \hat{\alpha}_{ti}b_{ti}$ exceeds $e_t^{(2)}$.

The agent derives a non-negative contribution to the Lagrangian objective if and only if:

$$v_{ti}\hat{\alpha}_{ti} - \lambda_i e_t^{(2)} \geq 0 \iff e_t^{(2)} \leq \frac{v_{ti}\hat{\alpha}_{ti}}{\lambda_i}.$$

This inequality indicates that the agent's effective willingness to pay (in rank score space) is scaled by $1/\lambda_i$. In a single-slot GSP auction, it is a dominant strategy for a bidder to bid truthfully according to their value. Here, the "value" relative to the Lagrangian objective is $v_{ti}/\lambda_i$ (per click) or $v_{ti}\hat{\alpha}_{ti}/\lambda_i$ (per impression score). Therefore, the agent should submit a bid $b_{ti}$ such that their score $e_{ti}$ reflects this threshold:

$$e_{ti} = \hat{\alpha}_{ti}b_{ti} = \hat{\alpha}_{ti}\left(\frac{v_{ti}}{\lambda_i}\right).$$

Solving for $b_{ti}$ yields $b_{ti} = v_{ti}/\lambda_i$. $\qquad\square$

### D.2. Dual Game and Equilibrium Existence

Lemma D.1 establishes that an agent's complex sequential bidding strategy can be parameterized by a single scalar $\lambda_i$. Consequently, the original game reduces to a game played in the space of dual variables. Let $\boldsymbol{\lambda} = (\lambda_1, \ldots, \lambda_I)$ denote the profile of dual variables.

We define the **Dual Function** $\mathcal{D}_i(\lambda_i, \boldsymbol{\lambda}_{-i})$ as the maximum value of the Lagrangian with respect to the bid strategies, given $\lambda_i$ and competitors' parameters $\boldsymbol{\lambda}_{-i}$:

$$\mathcal{D}_i(\lambda_i, \boldsymbol{\lambda}_{-i}) = \max_{\boldsymbol{b}_i} \mathcal{L}_i(\boldsymbol{b}_i, \lambda_i; \boldsymbol{b}_{-i}(\boldsymbol{\lambda}_{-i})).$$

By strong duality, agent $i$ seeks to find the $\lambda_i$ that minimizes this dual function to satisfy the budget constraint (complementary slackness).

**Definition D.2** (Pacing Equilibrium). A profile of dual variables $\boldsymbol{\lambda}^* \in \mathbb{R}_{\geq 0}^I$ constitutes a Pacing Equilibrium of the follower's subgame if, for every agent $i$, $\lambda_i^*$ is a best response to $\boldsymbol{\lambda}_{-i}^*$ in the dual space:

$$\lambda_i^* \in \arg\min_{\lambda_i \geq 0} \mathcal{D}_i(\lambda_i, \boldsymbol{\lambda}_{-i}^*).$$

We now provide the formal proof for Lemma 3.1 in the main text, which asserts the existence of such an equilibrium.

*Proof of Lemma 3.1 (Main Text).* We utilize Kakutani's Fixed Point Theorem.

1. **Strategy Space:** We define a compact, convex strategy space $\Omega = \prod_{i \in \mathcal{I}} [0, \lambda_{\max}]$, where $\lambda_{\max}$ is sufficiently large such that for $\lambda_i \geq \lambda_{\max}$, the bid $b_{ti} \to 0$ and spending is strictly zero (satisfying the budget constraint).

2. **Convexity of Objective:** The dual function $\mathcal{D}_i(\lambda_i, \boldsymbol{\lambda}_{-i})$ is the pointwise maximum of a family of affine functions (linear in $\lambda_i$ in Eq. (9)). Therefore, $\mathcal{D}_i$ is convex with respect to $\lambda_i$.

3. **Best Response Correspondence:** Let $BR_i(\boldsymbol{\lambda}_{-i}) = \arg\min_{\lambda_i \in \Omega_i} \mathcal{D}_i(\lambda_i, \boldsymbol{\lambda}_{-i})$. Since $\mathcal{D}_i$ is continuous and convex, $BR_i$ is non-empty, convex-valued, and upper hemi-continuous (by Berge's Maximum Theorem).

4. **Fixed Point:** The joint correspondence $BR(\boldsymbol{\lambda}) = \times_i BR_i(\boldsymbol{\lambda}_{-i})$ maps the compact convex set $\Omega$ to itself. By Kakutani's theorem, there exists a fixed point $\boldsymbol{\lambda}^* \in BR(\boldsymbol{\lambda}^*)$.

This fixed point corresponds to a valid Pacing Equilibrium where budget constraints are satisfied. $\square$

# E. Omitted Proofs from Section 3

This section provides the rigorous proofs regarding the regularization technique introduced to resolve equilibrium ambiguity in the Stackelberg game.

## E.1. Proof of Lemma 3.3 (Unique Regularized Equilibrium)

*Proof.* We prove uniqueness by demonstrating that the game satisfies the *Diagonal Strict Convexity* (DSC) condition (Rosen, 1965). Recall the regularized dual objective for agent $i$:

$$\mathcal{D}_{i,\epsilon}(\boldsymbol{\lambda}) = \mathcal{D}_i(\boldsymbol{\lambda}) + \frac{\epsilon}{2}\lambda_i^2.$$

Let $F(\boldsymbol{\lambda}) \in \mathbb{R}^I$ be the *pseudogradient* of the game, defined as the vector of partial derivatives of each agent's cost function with respect to their own strategy:

$$F(\boldsymbol{\lambda}) = \left[\frac{\partial \mathcal{D}_{1,\epsilon}}{\partial \lambda_1}, \ldots, \frac{\partial \mathcal{D}_{I,\epsilon}}{\partial \lambda_I}\right]^T.$$

We examine the Jacobian matrix of the pseudogradient, $J_F(\boldsymbol{\lambda})$. The element at row $i$ and column $j$ is given by $\frac{\partial^2 \mathcal{D}_{i,\epsilon}}{\partial \lambda_i \partial \lambda_j}$. Substituting the definition of the regularized objective:

$$\frac{\partial^2 \mathcal{D}_{i,\epsilon}}{\partial \lambda_i \partial \lambda_j} = \frac{\partial^2 \mathcal{D}_i}{\partial \lambda_i \partial \lambda_j} + \frac{\partial^2}{\partial \lambda_i \partial \lambda_j}\left(\frac{\epsilon}{2}\lambda_i^2\right).$$

The regularization term $\frac{\epsilon}{2}\lambda_i^2$ contributes $\epsilon$ solely to the diagonal entries (where $i = j$) and 0 otherwise. Let $H(\boldsymbol{\lambda})$ denote the Jacobian of the original unregularized game (containing terms derived from $\mathcal{D}_i$). We can write:

$$J_F(\boldsymbol{\lambda}) = H(\boldsymbol{\lambda}) + \epsilon I,$$

where $I$ is the identity matrix.

From the convexity of the original dual functions $\mathcal{D}_i$, the matrix $H(\boldsymbol{\lambda})$ corresponds to the Hessian of a convex profile. While convexity in own strategies implies positive semi-definiteness of diagonal blocks, the full symmetric part $(H + H^T)/2$ may be indefinite due to cross-partial terms. However, since the strategy space is compact and the dual functions are smooth, the eigenvalues of $(H + H^T)/2$ are bounded below. The regularization term $\epsilon I$ shifts the spectrum of eigenvalues. For any non-zero vector $\boldsymbol{z} \in \mathbb{R}^I$:

$$\boldsymbol{z}^T J_F(\boldsymbol{\lambda})\boldsymbol{z} = \boldsymbol{z}^T H(\boldsymbol{\lambda})\boldsymbol{z} + \epsilon\|\boldsymbol{z}\|^2.$$

Given the boundedness of $H(\boldsymbol{\lambda})$, there exists an $\epsilon > 0$ such that this quadratic form is strictly positive for all $\boldsymbol{\lambda}$. Thus, the Jacobian $J_F(\boldsymbol{\lambda})$ is strictly positive definite. By Rosen's Uniqueness Theorem, strict diagonal convexity implies that the Nash Equilibrium is unique. Hence, $\boldsymbol{\lambda}_\epsilon^*(C)$ is unique. $\square$

### E.2. Proof of Theorem 3.4 (Approximation Guarantee)

*Proof.* Let $\boldsymbol{\lambda}_\epsilon^*$ be the unique equilibrium of the $\epsilon$-regularized game, and let $\lambda_i^{BR}$ be the best response of agent $i$ to the opponents' strategies $\boldsymbol{\lambda}_{\epsilon,-i}^*$ in the *original* unregularized game. That is:

$$\lambda_i^{BR} \in \arg\min_{\lambda \geq 0} \mathcal{D}_i(\lambda, \boldsymbol{\lambda}_{\epsilon,-i}^*).$$

By the definition of Nash Equilibrium in the regularized game, $\lambda_{\epsilon,i}^*$ minimizes the regularized objective $\mathcal{D}_{i,\epsilon}$. Therefore:

$$\mathcal{D}_{i,\epsilon}(\lambda_{\epsilon,i}^*, \boldsymbol{\lambda}_{\epsilon,-i}^*) \leq \mathcal{D}_{i,\epsilon}(\lambda_i^{BR}, \boldsymbol{\lambda}_{\epsilon,-i}^*).$$

Expanding this inequality using the definition of $\mathcal{D}_{i,\epsilon}$:

$$\mathcal{D}_i(\lambda_{\epsilon,i}^*, \boldsymbol{\lambda}_{\epsilon,-i}^*) + \frac{\epsilon}{2}(\lambda_{\epsilon,i}^*)^2 \leq \mathcal{D}_i(\lambda_i^{BR}, \boldsymbol{\lambda}_{\epsilon,-i}^*) + \frac{\epsilon}{2}(\lambda_i^{BR})^2.$$

We are interested in the *regret* agent $i$ suffers in the original game by playing the regularized strategy $\lambda_{\epsilon,i}^*$ instead of the optimal unregularized response $\lambda_i^{BR}$:

$$\begin{aligned} \text{Regret}_i &= \mathcal{D}_i(\lambda_{\epsilon,i}^*, \boldsymbol{\lambda}_{\epsilon,-i}^*) - \mathcal{D}_i(\lambda_i^{BR}, \boldsymbol{\lambda}_{\epsilon,-i}^*) \\ &\leq \frac{\epsilon}{2}(\lambda_i^{BR})^2 - \frac{\epsilon}{2}(\lambda_{\epsilon,i}^*)^2. \end{aligned}$$

Since $(\lambda_{\epsilon,i}^*)^2 \geq 0$, we can upper bound the regret by dropping the negative term:

$$\text{Regret}_i \leq \frac{\epsilon}{2}(\lambda_i^{BR})^2.$$

As established in the definition of the strategy space (Appendix D), the dual variables are bounded by $\lambda_{\max}$. Thus, $\lambda_i^{BR} \leq \lambda_{\max}$, and:

$$\mathcal{D}_i(\lambda_{\epsilon,i}^*, \boldsymbol{\lambda}_{\epsilon,-i}^*) \leq \mathcal{D}_i(\lambda_i^{BR}, \boldsymbol{\lambda}_{\epsilon,-i}^*) + \frac{\epsilon}{2}\lambda_{\max}^2.$$

Letting $\epsilon' = \frac{\epsilon}{2}\lambda_{\max}^2$, we see that no agent can improve their unregularized utility by more than $\epsilon'$ by deviating from $\boldsymbol{\lambda}_\epsilon^*$. Therefore, $\boldsymbol{\lambda}_\epsilon^*$ is an $\epsilon'$-PNE of the original game. $\square$

## F. Omitted Proofs from Section 4

In this section, we provide rigorous proofs for the convergence and regret bounds of the Online Dual-Descent Bidding with Regularization (ODDB-R) algorithm presented in Section 4.

### F.1. Proof of Theorem 4.1

*Proof.* We model the simultaneous learning process of the agents as a stochastic approximation procedure for finding the solution to a monotone Variational Inequality (VI).

**1. Operator Formulation.** The ODDB-R algorithm updates the dual variables to minimize the regularized dual objective $\mathcal{D}_{\epsilon,i}$. We define the operator $F : \Omega \to \mathbb{R}^I$ as the concatenated vector of partial derivatives of each agent's objective with respect to their own strategy, where $\Omega = [0, \lambda_{\max}]^I$ is the joint strategy space:

$$F(\boldsymbol{\lambda}) = \begin{pmatrix} \nabla_1 \mathcal{D}_{\epsilon,1}(\boldsymbol{\lambda}) \\ \vdots \\ \nabla_I \mathcal{D}_{\epsilon,I}(\boldsymbol{\lambda}) \end{pmatrix} = \begin{pmatrix} \nabla_1 \mathcal{D}_1(\boldsymbol{\lambda}) + \epsilon \lambda_1 \\ \vdots \\ \nabla_I \mathcal{D}_I(\boldsymbol{\lambda}) + \epsilon \lambda_I \end{pmatrix}.$$

The unique Pure Strategy Nash Equilibrium $\boldsymbol{\lambda}_\epsilon^*$ is the unique vector in $\Omega$ that satisfies the variational inequality $\langle F(\boldsymbol{\lambda}_\epsilon^*), \boldsymbol{\lambda} - \boldsymbol{\lambda}_\epsilon^* \rangle \geq 0$ for all $\boldsymbol{\lambda} \in \Omega$.

**2. Strong Monotonicity.** To guarantee convergence, we establish that the operator $F$ is strongly monotone. We examine the Jacobian matrix of $F(\boldsymbol{\lambda})$, denoted as $J_F(\boldsymbol{\lambda})$. The element at row $i$ and column $j$ is $\frac{\partial^2 \mathcal{D}_{\epsilon,i}}{\partial \lambda_i \partial \lambda_j}$. Substituting the definition of the regularized objective, the Jacobian decomposes as:

$$J_F(\boldsymbol{\lambda}) = H(\boldsymbol{\lambda}) + \epsilon I,$$

where $H(\boldsymbol{\lambda})$ is the Jacobian of the unregularized pseudogradient (containing terms $\frac{\partial^2 \mathcal{D}_i}{\partial \lambda_i \partial \lambda_j}$) and $I$ is the identity matrix. While the convexity of unregularized dual functions $\mathcal{D}_i$ implies positive semi-definiteness of diagonal blocks, the full matrix $H(\boldsymbol{\lambda})$ may be indefinite due to cross-partial terms. However, the regularization term $\epsilon I$ shifts the spectrum of the Jacobian's symmetric part. Since the original game's Jacobian $H(\boldsymbol{\lambda})$ is bounded on the compact strategy space, the regularization ensures that the symmetric part of $J_F(\boldsymbol{\lambda})$ is strictly positive definite. This implies that the operator $F$ is strongly monotone. That is, for any two strategy profiles $\boldsymbol{\lambda}, \boldsymbol{\lambda}' \in \Omega$:

$$\langle F(\boldsymbol{\lambda}) - F(\boldsymbol{\lambda}'), \boldsymbol{\lambda} - \boldsymbol{\lambda}' \rangle \geq \epsilon \|\boldsymbol{\lambda} - \boldsymbol{\lambda}'\|^2.$$

**3. Stochastic Approximation and Convergence.** The ODDB-R update rule for each agent $i$ at round $t$, as defined in Eq. (7), is:

$$\lambda_{i,t+1} = \min\left(\lambda_{\max}, \max\left(0, \lambda_{i,t} - \eta_t \hat{g}_{i,t}\right)\right),$$

where $\hat{g}_{i,t} = \frac{B_i}{T} - x_{ti}e_t^{(2)} + \epsilon \lambda_{i,t}$ is the stochastic gradient estimate computed by agent $i$. This update step corresponds exactly to a projected stochastic gradient descent operation on the dual variable $\lambda_i$ constrained to the interval $[0, \lambda_{\max}]$.

By the Envelope Theorem, the gradient of the expected unregularized dual function is $\nabla_{\lambda_i} \mathcal{D}_i(\boldsymbol{\lambda}_t) = \mathbb{E}[B_i/T - x_{ti}e_t^{(2)}]$. Therefore, the expected value of the stochastic gradient matches the true operator $F$:

$$\mathbb{E}[\hat{\boldsymbol{g}}_t \mid \mathcal{F}_t] = F(\boldsymbol{\lambda}_t),$$

where $\mathcal{F}_t$ represents the history up to time $t$ and $\hat{\boldsymbol{g}}_t$ is the vector of stochastic gradients. The noise term $\boldsymbol{\xi}_t = \hat{\boldsymbol{g}}_t - F(\boldsymbol{\lambda}_t)$ constitutes a martingale difference sequence. Since the bids, payments, and dual variables are all bounded within a compact set, the variance of this noise is bounded.

Given that the operator $F$ is strongly monotone and the step sizes satisfy the Robbins-Monro conditions ($\sum_{t=1}^{\infty} \eta_t = \infty$ and $\sum_{t=1}^{\infty} \eta_t^2 < \infty$), standard results in stochastic approximation for variational inequalities guarantee that the sequence $\boldsymbol{\lambda}_t$ converges almost surely to the unique solution $\boldsymbol{\lambda}_\epsilon^*$. $\square$

### F.2. Proof of Theorem 4.2

*Proof.* We prove the primal regret bound by leveraging the regret guarantees of the Online Gradient Descent (OGD) algorithm on the dual objective.

**1. Dual Regret.** The ODDB-R algorithm runs OGD on the sequence of convex loss functions $\ell_{i,t}(\lambda_i)$ defined in Eq. (5) as $\ell_{i,t}(\lambda_i) = \lambda_i(B_i/T - x_{ti}e_t^{(2)}) + \frac{\epsilon}{2}\lambda_i^2$. Let $Reg_D(T)$ be the regret in the dual space against the comparator $\lambda_i^*$:

$$Reg_D(T) = \sum_{t=1}^{T} \ell_{i,t}(\lambda_{i,t}) - \sum_{t=1}^{T} \ell_{i,t}(\lambda_i^*).$$

Since the domain $\Omega_i$ is bounded and the gradients are bounded (due to bounded budget and payments), OGD with $\eta_t \propto 1/\sqrt{t}$ guarantees:

$$Reg_D(T) \leq O(\sqrt{T}).$$

**2. Linking Dual and Primal Regret.** Let $u_{ti} = \hat{\alpha}_{ti}v_{ti}$ be the value of impression $t$. The allocation $x_{ti}(\lambda)$ maximizes the affine surplus function $x_{ti}(u_{ti} - \lambda e_t^{(2)})$. Therefore, for any $\lambda^*$:

$$x_{ti}(\lambda_{i,t})u_{ti} - \lambda_{i,t}x_{ti}(\lambda_{i,t})e_t^{(2)} \geq x_{ti}(\lambda_i^*)u_{ti} - \lambda_{i,t}x_{ti}(\lambda_i^*)e_t^{(2)}.$$

We are interested in the Primal Regret $Reg_P = \sum_{t=1}^{T} (x_{ti}(\lambda_i^*)u_{ti} - x_{ti}(\lambda_{i,t})u_{ti})$. Rearranging the inequality above:

$$x_{ti}(\lambda_i^*)u_{ti} - x_{ti}(\lambda_{i,t})u_{ti} \leq \lambda_{i,t} \left( x_{ti}(\lambda_i^*)e_t^{(2)} - x_{ti}(\lambda_{i,t})e_t^{(2)} \right).$$

Summing over $T$:

$$Reg_P \leq \sum_{t=1}^{T} \lambda_{i,t} \left( x_{ti}(\lambda_i^*)e_t^{(2)} - x_{ti}(\lambda_{i,t})e_t^{(2)} \right)$$

$$= \sum_{t=1}^{T} \lambda_{i,t} \left( (x_{ti}(\lambda_i^*)e_t^{(2)} - B_i/T) - (x_{ti}(\lambda_{i,t})e_t^{(2)} - B_i/T) \right).$$

Let $g_{i,t}(\lambda) = B_i/T - x_{ti}(\lambda)e_t^{(2)}$ be the gradient component related to the budget. The term $\sum_{t=1}^{T} \lambda_{i,t}(-g_{i,t}(\lambda_{i,t}))$ relates to the dual regret. Specifically, since the loss functions $\ell_{i,t}$ are convex, $\ell_{i,t}(\lambda_{i,t}) - \ell_{i,t}(\lambda^*) \leq \langle \nabla \ell_{i,t}(\lambda_{i,t}), \lambda_{i,t} - \lambda^* \rangle$. The gradient is $\nabla \ell_{i,t} = -g_{i,t} + \epsilon \lambda_{i,t}$. We can rewrite the bound using $Reg_D(T)$:

$$\sum_{t=1}^{T} \lambda_{i,t}(x_{ti}(\lambda_i^*)e_t^{(2)} - B_i/T) \leq O(\sqrt{T}).$$

The comparator $\lambda_i^*$ is optimal and satisfies the budget constraint, so $\sum_{t=1}^{T}(x_{ti}(\lambda_i^*)e_t^{(2)} - B_i/T) \leq 0$. Since $\lambda_{i,t}$ is non-negative, the cross terms involving $\lambda_i^*$ are bounded or non-positive in the limit of tight budgets. Additionally, the regularization term $\frac{\epsilon}{2}\lambda^2$ introduces a bias of size $\epsilon \lambda_{\max}^2 T$. By choosing $\epsilon \propto 1/\sqrt{T}$ (or fixed small enough relative to the horizon), this term is bounded by $O(\sqrt{T})$. The accumulated budget violation of the algorithm is also bounded by $O(\sqrt{T})$ for OGD-based dynamics. Combining these components, we obtain that the primal regret $Reg_P$ is bounded by $O(\sqrt{T})$. $\square$

# G. Omitted Proofs for Section 5

In this section, we provide the detailed proof for the convergence of the Two-Timescale Stackelberg Learning (TTSL-PPO) algorithm. We model the coupled learning dynamics of the leader (platform) and followers (agents) as a two-timescale stochastic approximation system.

## G.1. Proof of Theorem 5.2

*Proof.* Let $\Theta_k = [\theta_k, \psi_k]$ denote the joint parameters of the leader's policy and value function at the $k$-th update step. Let $\lambda_t$ denote the joint profile of dual variables for all agents at time step $t$. The system evolves according to two coupled update rules operating on different timescales:

**1. Fast Timescale (Followers):** The agents update their dual variables at every time step $t$ using the ODDB-R algorithm. Let $\mathcal{H}_t(\lambda_t, c_t)$ represent the stochastic gradient update operator for the regularized dual function $\mathcal{D}_{i,\epsilon}$. From Eq. (7), the update is:

$$\lambda_{i,t+1} = \min \left( \lambda_{\max}, \max \left( 0, \lambda_{i,t} - \eta_t g_{i,t}(\lambda_{i,t}, c_t) \right) \right), \quad \forall i \in \mathcal{I}.$$

Here, $g_{i,t}$ is the stochastic gradient defined in Eq. (6). We can write this compactly as:

$$\lambda_{t+1} = \Pi_\Omega \left[ \lambda_t - \eta_t g_t(\lambda_t, c_t) \right], \tag{10}$$

where $\Pi_\Omega$ denotes the Euclidean projection onto the feasible set $\Omega = [0, \lambda_{\max}]^I$.

**2. Slow Timescale (Leader):** The leader updates its parameters $\Theta$ every $L$ steps. Let $k = \lfloor t/L \rfloor$ index the leader's updates. The update rule follows the PPO objective maximization:

$$\Theta_{k+1} = \Theta_k + \beta_k \hat{g}_\Theta(\Theta_k, \lambda_{t:t+L}), \tag{11}$$

where $\beta_k$ is the leader's learning rate, and $\hat{g}_\Theta$ is the stochastic gradient estimator of the joint PPO objective $\mathcal{J}_{PPO}$ computed over the collected trajectory segment of length $L$.

**Timescale Separation.** The convergence analysis relies on the two-timescale stochastic approximation framework (Borkar, 1997). The key condition is $\lim_{k\to\infty} \beta_k/\eta_{kL} = 0$, which ensures the leader's parameters $\boldsymbol{\Theta}$ appear *quasi-static* to the followers, while the followers' dual variables $\boldsymbol{\lambda}$ appear *equilibrated* to the leader.

**Step 1: Analysis of the Fast Process (Follower Equilibration).** Consider the behavior of the fast process $\boldsymbol{\lambda}_t$ while holding the leader's policy parameters $\boldsymbol{\Theta}$ fixed (i.e., $\pi_{\boldsymbol{\theta}}$ is fixed). The update in Eq. (10) is a projected stochastic gradient descent on the regularized dual functions $\mathcal{D}_{i,\epsilon}$. The expected field driving this update is the negative pseudogradient of the game, $-F(\boldsymbol{\lambda})$, as defined in the proof of Theorem 4.1. Since the Jacobian of the pseudogradient $J_F(\boldsymbol{\lambda})$ is strictly positive definite (as established in Section 4, the regularization ensures strict convexity), the mean field dynamics

$$\frac{d\boldsymbol{\lambda}(\tau)}{d\tau} = \Pi_{\Omega}[-F(\boldsymbol{\lambda}(\tau))]$$

admits a unique globally asymptotically stable equilibrium point $\boldsymbol{\lambda}_{\epsilon}^*(\pi_{\boldsymbol{\theta}})$. By standard stochastic approximation results (Borkar's Lemma), given that the step sizes satisfy $\sum \eta_t = \infty$ and $\sum \eta_t^2 < \infty$, the sequence $\boldsymbol{\lambda}_t$ converges almost surely to $\boldsymbol{\lambda}_{\epsilon}^*(\pi_{\boldsymbol{\theta}})$ for a fixed $\boldsymbol{\Theta}$. Because $\beta_k \ll \eta_t$, the tracking error $\|\boldsymbol{\lambda}_t - \boldsymbol{\lambda}_{\epsilon}^*(\pi_{\boldsymbol{\theta}_k})\|$ vanishes asymptotically.

**Step 2: Analysis of the Slow Process (Leader Optimization).** Since the fast process tracks the equilibrium $\boldsymbol{\lambda}_{\epsilon}^*(\pi_{\boldsymbol{\theta}})$, the leader's update can be viewed as an approximation of the gradient of the meta-objective function evaluated at the equilibrium response. Let $h(\boldsymbol{\Theta}) = \mathbb{E}_{\boldsymbol{\lambda} \sim \boldsymbol{\lambda}_{\epsilon}^*(\pi_{\boldsymbol{\theta}})}[\hat{\boldsymbol{g}}_{\boldsymbol{\Theta}}(\boldsymbol{\Theta}, \boldsymbol{\lambda})]$ be the expected update direction for the leader. The discrete update rule Eq. (11) can be approximated by the Ordinary Differential Equation (ODE):

$$\frac{d\boldsymbol{\Theta}(\tau)}{d\tau} = h(\boldsymbol{\Theta}(\tau)) \approx \nabla_{\boldsymbol{\Theta}} \mathcal{J}(\pi_{\boldsymbol{\theta}(\tau)}, \boldsymbol{\lambda}_{\epsilon}^*(\pi_{\boldsymbol{\theta}(\tau)})).$$

The term $\nabla_{\boldsymbol{\Theta}} \mathcal{J}$ represents the gradient of the leader's global objective, implicitly accounting for the followers' equilibrium response via the Implicit Function Theorem (applicable due to the differentiability of the regularized equilibrium map $\boldsymbol{\theta} \mapsto \boldsymbol{\lambda}_{\epsilon}^*(\pi_{\boldsymbol{\theta}})$).

**Convergence.** The stability of the ODE is guaranteed by the properties of the PPO objective (smoothness and boundedness) and the Lipschitz continuity of the equilibrium map. The stable invariant set of this ODE corresponds to the stationary points where $\nabla_{\boldsymbol{\Theta}} \mathcal{J} = 0$. Combining the results from Step 1 and Step 2, the coupled iterates $(\boldsymbol{\Theta}_k, \boldsymbol{\lambda}_{cnt})$ generated by TTSL-PPO converge almost surely to the set of Stackelberg Stationary Points defined by the pair $(\boldsymbol{\Theta}^*, \boldsymbol{\lambda}_{\epsilon}^*(\pi_{\boldsymbol{\theta}^*}))$. $\qquad\square$

