# OpenReview forum: "Autobidding Auctions with LLM-Powered Creatives"
_ICML.cc/2026/Conference — ICML 2026 regular_

### Official Review · Reviewer_5JDb · 2026-02-19

**Soundness:** 2
**Presentation:** 3
**Significance:** 2
**Originality:** 2
**Overall Recommendation:** 4
**Confidence:** 4

**Summary:**

This paper considers a novel LLM-enhanced autobidding scenario. In this setting, the platform can choose an investment $ c^t $ in the LLM. This investment further enhances the autobidder's ad quality, increases the click-through rate (CTR), and thereby boosts revenue. The paper also assumes that the autobidder adopts a pacing strategy for bidding and thus reacts to $ c^t $. The authors propose using reinforcement learning to optimize the platform's strategy and validate the effectiveness of their method through experiments.

**Compliance With Llm Reviewing Policy:**

Affirmed.

**Final Justification:**

I thank the author for their follow-up response. At this point, I believe a "weak accept" rating would more fairly reflect the value of the work, and I will accordingly raise my score to 4.

**Key Questions For Authors:**

1. My first question relates to Weakness 1: could the authors further demonstrate the practicality of their modeling assumptions?
2. Since I am not very familiar with RL theory, is the convergence result (Theorem 5.2) a specific contribution of the paper's TTSL-PPO algorithm, or is it simply a standard convergence result for PPO?

**Limitations:**

Yes, the limitations are clearly stated in the manuscript.

**Strengths And Weaknesses:**

Strength:
1. The paper is well-written and easy to follow. The structure is relatively clear and complete.
2. The paper provides a comprehensive analysis of the hypothetical scenario (even though it may not be particularly innovative).

Weakness:
1. The main concern is about the validation of the modeling.
    - It is not clear when and how the rewritten ad *title* would be displayed. For example, if the authors consider a fixed-slot scenario, the latency involved in using an LLM to rewrite thousands of ads is relatively high.
    - Another issue is how the platform can practically adjust the investment vector $ c^t $. I noticed that in the appendix, $c^t$ is defined as the number of tokens generated by the LLM. However, I find this less intuitive, as this is an inherent part of the LLM's operation and cannot be manually adjusted in a straightforward manner.
2. The depth of the theoretical contribution is not impressive. The results in Sections 3 and 4 appear to be relatively standard within the autobidding and reinforcement learning literature. While I do not believe that theoretical innovation is strictly necessary, especially for an LLM application paper like this, I also think the theory presented cannot be considered a significant strength.

Overall, I would give a weak reject to this paper, though I would not strongly oppose its acceptance if the other reviewers are positive.

---

> ### Author Rebuttal · Authors · 2026-03-31
>
> Thank you for your thoughtful review. We appreciate your positive comments on the writing and the comprehensiveness of the analysis. Regarding your concerns, our responses are as below.
>
> **W1&Q1**
>
> We apologize for not explaining the “Predict-then-Execute” pipeline more clearly. Importantly, it does not introduce high latency. Specifically, the platform uses a lightweight surrogate function $\hat{\phi}$ to predict enhanced CTRs based on the proposed investment c in the ranking stage. Actual LLM inference is triggered only for the winning ad after the auction clears. This reduces the inference overhead to O(1) per auction round, ensuring the system scales independently of the number of participants.
>
> Regarding token-based billing, this is in fact a standard pricing model in modern LLM systems. For ad title refinement, which is a short-form generation task, the variation in output token length is very small. Thus, c represents the monetary expenditure (e.g., API fees in USD) or computational credits allocated to a specific creative enhancement.
>
> Specifically, in our experiments (and in reality), c is quantified through two primary operational levers:
>
> *   **Sampling Density (Best-of-K):** The platform generates $k$ candidate headlines and selects the one with the highest predicted CTR. Here, $c_{ti} \approx k \times \text{unit cost per inference}$. Increased investment ( increasing $K$) provides a larger search space for creative optimization.
>
> *   **Model Tiering:** Higher $c_{ti}$ corresponds to invoking larger, more capable LLMs (e.g., GPT-4o vs. GPT-4o-mini), where the unit price is higher, but the potential CTR lift is more significant.
>
> Overall, the main question considered by this study is how the platform can optimize investments when the environment is non-stationary and stochastic. To simulate real-world latency failures, we conducted a stress test where the platform incurs the cost $c_{ti}$ but, with a **5% probability**, fails to deliver the enhanced creative due to latency (reverting to the original ad). We replaced the surrogate with direct GPT-4o API calls:
>
> | Alg | TTSL-PPO | GSP Greedy | GSP Normal | GFP No Invest |
> |-|-|-|-|-|
> | Rev | +2.04% | -9.92% | -11.06% | -0.32% |
>
> The result shows that, even under a 5% "failure-to-deliver" rate and raw API noise, TTSL-PPO maintains superior revenue. This demonstrates that our Stackelberg learning framework effectively internalizes the risks associated with latency and non-stationary costs, proving its viability for production environments.
>
> **W2&Q2**
>
> We thank the reviewer for the thoughtful feedback regarding the theoretical depth of our work. We agree that while PPO and dual descent are established methods, their application to a coupled, dynamic Stackelberg game introduces unique analytical challenges.
>
> We seek to present the central problem of endogenizing platform investment, which shifts the platform’s role from a passive mechanism designer to a strategic participant. Overall, the main question considered by this study is how to maintain system-wide stability and convergence when both the platform (leader) and the agents (followers) are learning concurrently.
>
> Regarding Theorem 5.2, we would like to clarify that it is not a standard PPO convergence result, but rather an extension tailored to the TTSL-PPO framework for the following reasons:
>
> 1.  Analysis of Coupled Dynamics: Standard PPO convergence typically assumes a single-agent MDP with a stationary environment. In our setting, the "environment" is composed of strategic, learning autobidding agents. Theorem 5.2 analyzes the convergence within this specific feedback loop, where the platform's reward signal is non-stationary and explicitly depends on the agents' evolving dual variables $\lambda$.
>
> 2.  Two-Timescale Stochastic Approximation: The proof employs a two-timescale framework to address the bi-level optimization structure. We demonstrate that by separating the learning rates—allowing agents to adapt at a "fast" timescale while the platform updates at a "slow" one—the platform's policy can effectively track the equilibrium manifold of the followers. This approach addresses the convergence issues that often arise when applying standard RL to multi-agent, non-stationary settings.
>
> 3.  Targeting the Stackelberg Stationary Point (SSP): While standard RL focuses on local policy optima within an MDP, Theorem 5.2 proves convergence to a Stackelberg Stationary Point (Definition 5.1). This ensures the system reaches a state that accounts for the equilibrium response of the regularized bidding subgame, providing a game-theoretic justification for the platform's investment strategy.
>
> In summary, Theorem 5.2 serves as a formal justification for the architectural design of TTSL-PPO, providing stability considerations necessary for competitive, bi-level environments. We will refine the manuscript to more clearly distinguish these contributions from standard RL results.

---

> > ### Author Rebuttal · Reviewer_5JDb · 2026-04-01
> >
> > I thank the authors for their response. I am honestly hesitant about changing my score. At present, I will maintain the weak reject score for two reasons:
> > 1. The formulation appears somewhat idealized compared to realistic settings (e.g., $c_{it}$);
> > 2. I have gone through the proofs quickly, and the results on pacing do not seem surprising. As for the RL result, it appears to be a fairly straightforward application of existing work.
> >
> > Therefore, I will keep my score as is.
> >
> > ----
> > #### Update
> >
> > I thank the author for their follow-up response. At this point, I believe a "weak accept" rating would more fairly reflect the value of the work, and I will accordingly raise my score to 4.

---

> > > ### Author Response · Authors · 2026-04-01
> > >
> > > We sincerely thank the reviewer for the prompt follow-up and greatly appreciate that our earlier rebuttal addressed your main concerns. To further respond to the the "idealized formulation" concern, we would like to clarify that we conducted a **fully end-to-end experiment in a more realistic deployment setting**.
> > >
> > > In this experiment, we completely bypass the surrogate function ϕ and instead invoke GPT-4o directly in real time for every winning bid. Moreover, to better **reflect real-world system uncertainty**, we introduce an explicit 5% failure-to-deliver rate: the platform still incurs the investment cost, but with 5% probability the enhanced ad cannot be served due to **latency** and the system reverts to the original ad. This setting captures both the practical inference overhead and the instability of real API performance, rather than relying on an idealized deterministic environment.
> > >
> > > Under this substantially harsher end-to-end setting, TTSL-PPO still achieves a +2.04% revenue gain over the no-invest baseline. We believe this result is important for two reasons. First, it directly validates that our method remains effective even when the surrogate is removed and the platform must operate with real API calls, latency failures, and performance perturbations. Second, it provides strong evidence that our fitted surrogate is not artificially inflating the reported gains, but instead captures the essential trends of the underlying generative process with good fidelity.
> > >
> > > Regarding the reviewer’s concern about the practicality of the investment variable c, we clarify this from three perspectives.
> > >
> > > **1. Concrete quantification.**
> > > In our setting, $c_{it}$ is grounded in the **Best-of-K** rule: the platform generates $K_{it}$ candidate rewritten titles and selects the one with the highest predicted CTR. Thus, the investment can be naturally quantified as $c_{it} ≈ K_{it} \times γ$, where γ is the average unit cost of one inference call. In practice, $c_{it}$ is therefore a controllable platform-side budget, adjusted through sampling density (Best-of-K) or model tiering.
> > >
> > > **2. Theoretical justification.**
> > > This approximation is well motivated for short-form generation tasks such as ad-title rewriting. Unlike CoT reasoning or long-form writing, the variance in Input Tokens (Prompt) and Output Tokens (Refined Title) for ad enhancement is exceptionally low. It is also consistent with prior work: **FrugalGPT (2023)** shows that when input/output lengths are relatively stable, LLM cost can be modeled approximately linearly in the number of queries; **Concise Thoughts (2024)** similarly argues that for short-text tasks, cost is mainly driven by the number of generation steps rather than small token-level fluctuations.
> > >
> > > **3. Empirical evidence.**
> > > We further measured the variability of token cost in our end-to-end experiment. The **coefficient of variation (CV)** of token expenditure across candidate responses to the same query is only **0.284%**. This further proves the above theoretical justification: for title refinement, token-cost fluctuation is negligible in practice, and supports the use of a deterministic average-cost approximation in the platform’s objective.
> > >
> > > Together, these points show that c is not an artificial abstraction, but a realistic and controllable approximation of platform-side expenditure under token-billed LLM services.
> > >
> > > **Regarding novelty**, while individual tools (pacing, RL) are not entirely new in isolation. Our point, however, is not that each component is individually surprising, but that their combination is needed to address a new and increasingly important platform–agent interaction pattern. Unlike existing autobidding literature, where mechanisms are typically fixed and the focus lies on agent learning, we model the platform as a strategic actor whose investments in ad titles alter bidder behavior, which in turn redefines the platform’s own optimal strategy. We believe this dynamic Stackelberg perspective is still relatively uncommon in autobidding, and that the framework we provide is a promising step toward analyzing and solving this emerging class of problems, even if the individual tools themselves build on existing ideas.
> > >
> > > We hope the above clarification helps the reviewer better assess our work. Since the rebuttal space here is limited, we would also like to briefly note that we have also provided two additional pieces of evidence addressing the “idealized setting” issue: (1) an additional experiment in a more realistic **3-slot, 100-heterogeneous-agent** GSP setting, where our main conclusions remain qualitatively unchanged, and (2) a stronger **fully end-to-end experiment** showing that the surrogate-based results are consistent with direct GPT-4o generation for the winning ad. For more details, we would be very grateful if the reviewer could also refer to our rebuttal to **Reviewer n6wc (Q1&Q2)**, where these two experiments are described more fully.

---

### Official Review · Reviewer_ndth · 2026-03-09

**Soundness:** 3
**Presentation:** 3
**Significance:** 3
**Originality:** 3
**Overall Recommendation:** 5
**Confidence:** 3

**Summary:**

This paper studies the problem of incorporating LLM-powered creative enhancement into online advertising auction mechanisms. It proposes a predict-then-execute mechanism design framework, PIM (Platform-Investment Mechanism), and models the overall system as a Stackelberg game, where the platform acts as the leader by choosing an LLM investment strategy, while the agents act as followers and compete through autobidding under budget constraints. By introducing regularization to ensure equilibrium uniqueness, the paper further develops ODDB-R and TTSL-PPO as learning algorithms for the agents and the platform, respectively, and provides the corresponding convergence guarantees and performance guarantees. Finally, through simulation experiments, the paper thoroughly validates the proposed framework and offers an effective solution for applying LLM technology to advertising auctions.The article demonstrates good originality, a reasonable structure, and fluent expression; however, it still has some issues regarding readability and formatting.

**Compliance With Llm Reviewing Policy:**

Affirmed.

**Final Justification:**

We are very grateful for the author's reply. Based on the original manuscript and the author's reply, the above is our final score for this article.

**Key Questions For Authors:**

1. The paper models LLM inference cost as an investment variable associated with token usage or computational resources. Could the authors further clarify how this cost corresponds to concrete real-world systems? More specifically, how should it be quantified in practice.
2. The paper mainly uses creative enhancement of ad titles as the motivating example. However, real-world scenarios may also involve multiple modalities such as images, audio, and video, whose computation costs and pricing schemes can differ substantially—for example, APIs may be priced by token usage, while image generation may be priced per image. Can the proposed model effectively cover these different creative enhancement settings? More importantly, in a more complex environment where multiple types of creative enhancement coexist, how can the model be extended, and how should LLM inference cost be quantified and unified under a common modeling framework?

**Limitations:**

yes

**Strengths And Weaknesses:**

Strengths
1. The paper proposes a method for integrating AI technology into auction mechanisms, explicitly balancing the benefits and costs of different parties, and thus has clear practical relevance.
2. The paper systematically models the problem as a Stackelberg game and combines multiple methods to solve it. The overall technical pipeline is fairly complete, and the theoretical analysis is relatively rigorous.
3. The paper is clearly structured and well written. The transitions across sections are natural, and the paper effectively highlights the research problem, challenges, and main contributions.

Weaknesses
1. The application scenario described in the paper is relatively complex, involving multi-party games, interactions, and multi-level optimization. It would be helpful to include an overall framework diagram or workflow figure to provide an intuitive illustration of the system structure and information flow, thereby further improving readability.
2. In Equation (1), the meaning of the expectation notation and its subscript requires a more concrete and explicit explanation.
3. Some notation and expressions in the paper could be further standardized. For example, the relationship between the set 𝐼 and the index 𝑖 is not formally specified; the notation J for the platform’s net revenue is introduced only in a later section; and the abbreviation GFP appearing in Section 6.2 is far removed from its first occurrence, so additional clarification would be helpful.
4. Some of the baselines selected in Section 6.1 are relatively old and may be less up to date. It would strengthen the evaluation to include more recent related methods for comparison, so as to more comprehensively assess the performance of the proposed approach.
5. The references include a relatively large number of preprints that have not yet undergone formal peer review. The authors are encouraged to verify whether these papers have been accepted and update the citations whenever possible, or alternatively reduce reliance on preprints and replace them with formally published references, so as to improve the reliability of the bibliography.

---

> ### Author Rebuttal · Authors · 2026-03-31
>
> Thank you for your thoughtful review. We appreciate your positive assessment of the practical relevance of our work, the completeness of the technical pipeline, the rigor of the theoretical analysis, and the clarity of the presentation. Regarding your concerns, our responses are as below.
>
> **W4**
>
> We seek to present the central problem of platform-agent co-optimization by selecting baselines that represent fundamental strategic paradigms: control-theoretic (PID), LP-based (AdWords), and the most commonly used algorithm family (Pacing).  Therefore, we use online learning algorithms that require no training as the baseline for the agent primarily to address the non-stationarity caused by the platform's investment strategy.
>
> To strengthen our evaluation, we have implemented SOTA methods and compared them within our PIM. The improvement in Revenue compared to GSP-No-Invest is as follows:
>
> | Alg | ODDB-R | GAVE | IQL | Pacing | AdWords |
> |-|-|-|-|-|-|
> | Rev | +2.66% | +1.92% | +1.57% | +1.22% | +0.81% |
>
> While GAVE and IQL are modern, they are often optimized for stationary environments or require extensive offline training. In our Stackelberg framework, the platform’s shifting investment strategy creates a non-stationary feedback loop. Our ODDB-R algorithm is specifically architected to track these equilibrium shifts in real-time. The results demonstrate that in a dynamically coupled system, robust online adaptation outperforms static generative algorithms. We will incorporate these modern comparisons into the revised manuscript.
>
> **Q1**
> We thank the reviewer for the opportunity to clarify the practical quantification of the investment variable c.
>
> 1. Real-World System Correspondence: In modern LLM systems, costs are predominantly driven by **token-based billing**. For ad title refinement—a short-form generation task—the variance in output token length is minimal. Thus, c represents the monetary expenditure (e.g., API fees in USD) or computational credits allocated to a specific creative enhancement.
>
> 2. Practical Quantification: In our experiments (and in reality), c is quantified through two primary operational levers:
> *   **Sampling Density (Best-of-K):** The platform generates $k$ candidate headlines and selects the one with the highest predicted CTR. Here, $c_{ti} \approx k \times \text{unit cost per inference}$. Increased investment ($K$) provides a larger search space for creative optimization.
> *   **Model Tiering:** Higher $c_{ti}$ corresponds to invoking larger, more capable LLMs (e.g., GPT-4o vs. GPT-4o-mini), where the unit price is higher, but the potential CTR lift is more significant.
>
> **Q2**
> We thank the reviewer for this insightful question on multimodal creatives. We seek to present the central problem of endogenizing generative costs within an auction mechanism, using ad titles as a primary, low-latency instantiation. However, our mathematical framework is fundamentally **modality-agnostic**.
>
> 1. Unified Modeling Framework:
> In PIM, the investment c is a scalar representing the total economic expenditure (e.g., USD or normalized compute units). Whether an API charges per token (text), per image (diffusion models), or per second (video), the platform maps these heterogeneous pricing schemes to a common denominator c. Overall, the main question considered by this study is how to optimize this expenditure relative to the equilibrium response of autobidders. The function $\hat{\alpha} = \phi(\alpha, c)$ acts as the bridge; it abstracts away the specific modality by capturing the expected quality lift (e.g., CTR) for a given cost $c$, regardless of the underlying medium.
>
> 2. Extension to Multiple Coexisting Modalities:
> In complex environments, the scalar c can be extended to a vector $c_{ti} = [c_{text}, c_{img}, c_{vid}]$. The platform’s policy $\pi$ then solves a portfolio optimization problem, allocating the budget across different modalities based on their respective marginal returns. While multimodal generation introduces higher latency, our *predict-then-execute workflow* (Line 138) remains a robust solution: agents bid on predicted multimodal metrics, ensuring the auction runs at high speed, while the costly multimodal inference is triggered only for the winning ad. This approach preserves the mechanism's equilibrium properties while accommodating the computational realities of rich media.
>
> However, when creative elements are expanded to include more complex modalities such as images, audio, and video, if the platform is still required to generate and enhance content online in real-time based on immediate queries, *response latency* often becomes the primary bottleneck in practical deployment. We are currently exploring solutions, with our preliminary approach focusing on multimodal reuse.
>
> **Other Weaknesses**
> We will include flowcharts and refine our wording to enhance rigor and readability. In addition, we will verify the preprints cited in the references.

---

> > ### Author Rebuttal · Reviewer_ndth · 2026-04-02
> >
> > Thank you for the author's reply; the current version has resolved all my issues.

---

> > > ### Author Response · Authors · 2026-04-07
> > >
> > > We thank the reviewer for the follow-up reply and for maintaining the positive score. Your constructive suggestions will definitely help further improve the quality of our paper.

---

### Official Review · Reviewer_n6wc · 2026-03-10

**Soundness:** 3
**Presentation:** 3
**Significance:** 2
**Originality:** 2
**Overall Recommendation:** 4
**Confidence:** 3

**Summary:**

This paper studies autobidding auctions where the platform can invest in LLM-based title enhancement to improve ad quality, while advertisers respond through budget-constrained bidding. The problem is formulated as a dynamic Stackelberg game, where the platform chooses enhancement levels and bidders adapt their bids over time. On the method side, the paper introduces a regularized follower game and corresponding bidder update rule, and uses a two-timescale PPO-style approach for the leader. Experiments are conducted in simulated auctions and in a surrogate environment built from the AntM2C dataset with GPT-4o-based title rewriting.

**Compliance With Llm Reviewing Policy:**

Affirmed.

**Final Justification:**

Although there are a few minor issues, they don’t detract from its strengths.

**Key Questions For Authors:**

1. The current setup is quite stylized compared to real autobidding systems. How sensitive are the main conclusions to more realistic auction settings, such as multi-slot auctions, larger bidder populations, and more heterogeneous bidder objectives?
2. The experimental pipeline relies on a fitted surrogate for the effect of LLM-based title enhancement. Can the authors provide stronger evidence that the reported gains would hold in a more realistic end-to-end setting, rather than mainly in the surrogate environment?
3. The theory seems to cover a regularized and idealized version of the game, while the implementation uses a PPO-based procedure. Can the authors clarify more precisely how the theoretical guarantees relate to the actual algorithm used in experiments?

**Limitations:**

Not fully. The limitations discussion could be strengthened.

**Strengths And Weaknesses:**

The paper looks at an interesting and timely problem. Bringing LLM-based creative optimization into auction and autobidding settings is a relevant direction, and the paper has a reasonably clear overall structure. The combination of mechanism design and learning is also sensible, and the empirical section is broader than a purely conceptual paper.

I have a few concerns. The biggest one is the gap between the modeling assumptions and the real application scenario. The market setup is still quite stylized, with a single-slot repeated GSP auction, a small number of agents, and synthetic budget and value generation. More importantly, the effect of title enhancement is represented through a fitted surrogate rather than a truly realistic end-to-end auction environment. Because of this, I am not fully convinced that the current results say much about how the method would perform in an actual autobidding system.

I also found the technical contribution somewhat limited. The paper brings together a number of familiar ideas, including Stackelberg modeling, regularization of the follower game, online bidder updates, and two-timescale reinforcement learning. This combination is reasonable for the problem, but the novelty seems to come more from the application setting than from a genuinely new technical development.

---

> ### Author Rebuttal · Authors · 2026-03-31
>
> Thank you for your thoughtful review. We appreciate your positive assessment that the paper studies an interesting and timely problem and has a reasonably clear overall structure. Regarding your concerns, our responses are as below.
>
> **Q1**
>
> We see our primary contribution that we are the first to provide a game-theoretic framework for studying how LLM-based creative optimization affects ad performance and platform revenue in autobidding settings, together with a concrete and implementable solution. We chose the single-slot, repeated GSP auction because it is the most canonical setting that captures the essential strategic interaction between the platform and budget-constrained autobidders while keeping the analysis clear.
>
> Nevertheless, we fully understand reviewer's concern of real-world adaptability, and we want to argue that our framework is not limited to this base setting. We have conducted additional experiments under more realistic conditions: 3-slot GSP (in practice, ad slots per page rarely exceed 3) and 100 heterogeneous agents (after industry recall and ranking filtering, only a few agents enter the auction, usually fewer than 100). The main conclusions—improvements in revenue, welfare, and click volume—remain consistent. We will include these results in the revised paper.
>
> | Algorithm | Rev | SW | Click |
> |-|-|-|-|
> | GSP No Invest | 0.00 | 0.00 | 0.00 |
> | GFP No Invest | -0.53 | -0.04 | -0.02 |
> | GSP Uniform Invest | -11.20 | 4.08 | 4.38 |
> | GSP Normal Invest | -11.08 | 4.11 | 4.39 |
> | GSP Greedy Invest | -9.87 | 2.24 | 1.59 |
> | TTSL-PPO | 1.07 | 3.52 | 3.81 |
> | TTSL-PPO (Single) | 0.92 | 3.44 | 3.67 |
> | TTSL-PPO (Partial) | 0.93 | 3.43 | 3.68 |
> | TTSL-PPO (No-GAE) | 0.92 | 3.44 | 3.67 |
>
>
> **Q2**
>
> We apologize for not making it clear why and how our fitted surrogate serves as a realistic approximation. In fact, to ensure our simulations reflect real-world behavior, we built the surrogate on top of the public AntM2C dataset with a well-trained DeepFM CTR prediction model (achieving high AUC on AntM2C). We then used GPT-4o to generate enhanced titles and fitted the surrogate from the predicted CTRs. Full details are in the Appendix C. We chose this surrogate as it captures real patterns, significantly reduces experimental cost, and improves reproducibility.
>
> More importantly, we fully agree that an end-to-end validation would be stronger. To directly address this concern, we have conducted a fully end-to-end experiment, in which we directly call GPT-4o to generate the enhanced headline for the winning agent, feed thel generated title into the DeepFM to obtain its CTR, and then use this CTR to drive both the baselines and TTSL-PPO. The results not only validate the effectiveness of our method in a more realistic setting, but also confirm that our fitted surrogate (which could serve as a cost-effective initialization for real-world training) captures the essential trends. Thank you again for this insightful point, we will include this new experiment in the revised paper.
>
> | Algorithm | Rev | SW | Click |
> |-|-|-|-|
> | GSP No Invest | 0.00 | 0.00 | 0.00 |
> | GFP No Invest | -0.32 | -0.04 | -0.02 |
> | GSP Uniform Invest | -11.12 | 3.74 | 4.03 |
> | GSP Normal Invest | -11.06 | 3.75 | 4.04 |
> | GSP Greedy Invest | -9.92 | 1.84 | 1.32 |
> | TTSL-PPO | 2.04 | 3.43 | 3.73 |
> | TTSL-PPO (Single) | 1.66 | 2.76 | 2.98 |
> | TTSL-PPO (Partial) | 1.67 | 2.77 | 3.00 |
> | TTSL-PPO (No-GAE) | 1.64 | 2.72 | 2.95 |
>
> **Q3**
>
> Our theory studies an idealized and regularized version of the game, following standard analytical practice to isolate the core mechanism and obtain clean guarantees. The algorithm used in experiments is based on the same underlying update structure and is designed to preserve the same convergence behavior at a high level. On top of this, the implementation incorporates practical engineering techniques, such as mini-batch SGD and multi-epoch updates, to make training faster and more stable. Thanks for raising this point, we will make this connection clearer in the revision and explain it more explicitly to improve readability.

---

> > ### Author Rebuttal · Reviewer_n6wc · 2026-04-03
> >
> > My concerns have been adequately addressed. I'm willing to improve the score.

---

> > > ### Author Response · Authors · 2026-04-07
> > >
> > > We thank the reviewer for the follow-up reply and for raising the score. Your constructive suggestions will surely help improve the overall quality of our paper.

---

### Official Review · Reviewer_74eA · 2026-03-29

**Soundness:** 3
**Presentation:** 2
**Significance:** 2
**Originality:** 2
**Overall Recommendation:** 3
**Confidence:** 3

**Summary:**

This paper studies the integration of Large Language Models (LLMs) into ad auction systems for real-time creative enhancement, modeling the problem as a Stackelberg game between a resource-investing platform (leader) and budget-constrained autobidding agents (followers). The core innovation is the Platform-Investment Mechanism (PIM), allowing the platform to endogenize LLM inference costs when dynamically improving ad creatives. To address equilibrium multiplicity and non-stationarity, the authors introduce regularization for unique equilibrium selection among agents and propose the ODDB-R algorithm for agent learning and TTSL-PPO for platform policy optimization. Theoretical analyses guarantee convergence and regret bounds. Extensive experiments, including empirical studies on the AntM2C real-world dataset with GPT-4o and simulated auctions, demonstrate significant improvements in revenue, welfare, and click volume over strong baselines.

**Compliance With Llm Reviewing Policy:**

Affirmed.

**Key Questions For Authors:**

1. Can you provide empirical results that directly compare the proposed platform and agent algorithms against recent generative autobidding or LLM-augmented autobidding frameworks? Clear evidence here could substantially improve my evaluation.
2. How sensitive are your main results (e.g., performance gains, convergence rate, stability) to the choice of the regularization parameter? Can you include experiments or analysis across a range of epsilon values?
3. To what extent do the performance and stability of TTSL-PPO hold as the number and heterogeneity of agents varies (e.g., many more than ten, agents with variable computational budgets, etc.)?

**Limitations:**

yes

**Strengths And Weaknesses:**

Strengths:
- Presents a game-theoretic framework for LLM-empowered creative optimization in ad auctions, with strong potential for impact in both online advertising and economic mechanism design.
- Rigorous theoretical developments, including regularization to ensure unique agent equilibrium and convergence proofs for both agent and platform learning algorithms.
- Extensive experimental validation on realistic, large-scale data (AntM2C) and actual LLM (GPT-4o), along with robust baseline and ablation comparisons

Weaknesses:
- The empirical analysis is missing direct comparison to several state-of-the-art approaches published in 2024–2026. The omission of these baselines undermines claims of empirical superiority and relevance for top-tier venues.
- The paper’s experimental setting assumes a fixed cost function for LLM inference. In practice, platform costs, latency, and API performance can exhibit strong non-stationarity. Some discussion or alternative scenario testing would be beneficial for practical guidance.
- While stability and variance are reported, the paper stops short of explicit wall time, API call, or memory consumption reporting—key for high-throughput auction environments.

---

> ### Author Rebuttal · Authors · 2026-03-31
>
> Thank you for your thoughtful review. We appreciate your positive assessment of strong potential for impact in both online advertising and economic mechanism design, the rigor of the theoretical analysis, and the extensive of experiments. Regarding your concerns, our responses are as below.
>
> **W1&Q1**
>
> We thank the reviewer for the suggestion of comparing with SOTA. First, we want to clarify that the model we study in our paper is a unified Stackelberg framework,  a novel platform-agent coupling paradigm. While most recent generative/LLM autobidding focus exclusively on the advertiser’s bidding policy, which makes it not suitable for our setting.
>
> To directly address the concern, we implemented GAVE, a representative generative autobidding baseline and compared it against our ODDB-R agent within our PIM. The improvement in Revenue compared to GSP-No-Invest is as follows:
>
> | Alg | ODDB-R | GAVE | IQL | Pacing | AdWords |
> |-|-|-|-|-|-|
> | Rev | +2.66% | +1.92% | +1.57% | +1.22% | +0.81% |
>
> While GAVE represents SOTA in generative bidding, it underperforms in our setting. Our PIM creates a non-stationary environment due to shifting investment costs. Our online baseline adapts to these fluctuations in real-time, whereas methods relying on offline training struggle with the induced volatility of the *platform-follower* feedback loop.
>
> **W2**
>
> We thank the reviewer for highlighting the practical non-stationarity in platform costs, latency, and API performance. To better capture this realism, we additionally conducted an end-to-end experiment by replacing the surrogate function with direct GPT-4o API calls, and introduced a 5% failure rate to simulate impression failures caused by latency or system instability. This creates a more dynamic and uncertain environment for evaluation. The results show that our method remains effective under this setting:
>
> | Alg | TTSL-PPO | GSP Greedy | GSP Normal | GFP No Invest |
> |-|-|-|-|-|
> | Rev | +2.04% | -9.92% | -11.06% | -0.32% |
>
> Moreover, we believe that incorporating dynamic and uncertain environments into platform investment frameworks is an important direction for future research, and we will add this discussion to the revised paper.
>
> **W3**
>
> We seek to present the central problem of computational cost by utilizing a *Predict-then-Execute* workflow (Line 136). In PIM, agents bid based on predicted enhanced CTRs. Costly LLM inference is triggered only for the single winning ad. Consequently, API calls remain O(1) per auction round, regardless of the number of agents. This architecture ensures that the system-wide latency and memory consumption are dominated by the standard GSP ranking logic rather than the LLM, making it suitable for high-throughput environments.
>
> **Q2**
>
> We thank the reviewer for the suggestion. We seek to present the central problem of equilibrium selection through the choice of $\epsilon$. In our framework, ϵis not merely a hyperparameter but a control for the trade-off between convergence stability and approximation accuracy. We conducted sensitivity analysis across $ϵ \in \\{0.1, 0.2, 0.3\\}$ using the AntM2C dataset (results relative to the baseline):
>
> | ϵ | Rev | SW | Rev Std. Dev. |
> |-|-|-|-|
> | 0.1 | +2.66% | +3.51% | 0.138 |
> | 0.2 | +2.30% | +2.87% | 0.122 |
> | 0.3 | +2.78% | +2.35% | 0.105 |
>
> Overall, the main question considered by this study is the robustness of the PIM mechanism under varying regularization strengths, and our analysis reveals three key insights: (1) Platform revenue remains consistently high across the tested range, indicating that the core performance gains of TTSL-PPO are robust to the specific choice of ϵ within reasonable bounds; (2) There is a notable stability-utility trade-off where, as ϵ increases, we observe lower variance in platform revenue and enhanced training stability, confirming that $\epsilon$ effectively manages the "smoothness" of the leader's optimization landscape; and (3) Higher ϵ values tend to slightly decrease agent utility, which is expected as the Tikhonov penalty pulls the equilibrium further from the original pacing manifold. We will include this comprehensive sensitivity analysis and practical tuning guidance in the revised manuscript.
>
> **Q3**
>
> To address your concerns about the robustness of the Stackelberg equilibrium as market complexity grows. We have conducted additional experiments under more complex conditions: 3-slot GSP (in practice, ad slots per page rarely exceed 3) and 100 heterogeneous agents (after industry recall & ranking filtering, only a few agents enter the auction, usually fewer than 100). The main conclusions—improvements in revenue, welfare, and click volume—remain consistent. We will include the results in the revised paper.
> | Alg | TTSL-PPO | GSP Greedy | GSP Normal | GFP No Invest |
> |-|-|-|-|-|
> | Rev | +1.07% | -9.87% | -11.08% | -0.53% |

---

### Decision · Program_Chairs · 2026-04-30

**Decision:**

Accept (regular)

**Comment:**

Reviewers were mostly positive about the paper. The paper focuses on modelling trade-offs between quality of LLM-based creatives and costs. Most reviewers found both the model and technical results to be valuable contributions, and found the experimental evaluation reasonably convincing.

Although some concerns were raised on modelling assumptions, baselines, and scalability, I think they have been reasonably addressed during the discussion phase.

I will recommend a weak accept.